# Androgen deprivation promotes neuroendocrine differentiation and angiogenesis through CREB-EZH2-TSP1 pathway in prostate cancers

Yan Zhang[1,2], Dayong Zheng[1,3], Ting Zhou [1,4], Haiping Song[1,5], Mohit Hulsurkar[1,14], Ning Su[1,6], Ying Liu[1,7], Zheng Wang[1], Long Shao[8], Michael Ittmann[8], Martin Gleave[9], Huanxing Han[10], Feng Xu [4], Wangjun Liao[11], Hongbo Wang[12] & Wenliang Li [1,13,14]

The incidence of aggressive neuroendocrine prostate cancers (NEPC) related to androgen-deprivation therapy (ADT) is rising. NEPC is still poorly understood, such as its neuroendocrine differentiation (NED) and angiogenic phenotypes. Here we reveal that NED and angiogenesis are molecularly connected through EZH2 (enhancer of zeste homolog 2). NED and angiogenesis are both regulated by ADT-activated CREB (cAMP response element-binding protein) that in turn enhances EZH2 activity. We also uncover anti-angiogenic factor TSP1 (thrombospondin-1, THBS1) as a direct target of EZH2 epigenetic repression. TSP1 is downregulated in advanced prostate cancer patient samples and negatively correlates with NE markers and EZH2. Furthermore, castration activates the CREB/EZH2 axis, concordantly affecting TSP1, angiogenesis and NE phenotypes in tumor xenografts. Notably, repressing CREB inhibits the CREB/EZH2 axis, tumor growth, NED, and angiogenesis in vivo. Taken together, we elucidate a new critical pathway, consisting of CREB/EZH2/TSP1, underlying ADT-enhanced NED and angiogenesis during prostate cancer progression.

[1] Texas Therapeutics Institute, Brown Foundation Institute of Molecular Medicine, University of Texas Health Science Center at Houston, Houston, TX 77030, USA. [2] Department of Anesthesiology, Union Hospital of Tongji Medical College, Huazhong University of Science and Technology, Wuhan, 430022, China. [3] Cancer Center, Integrated Hospital of Traditional Chinese Medicine, Southern Medical University, Guangzhou, 510513, China. [4] Department of Pharmacy, Fengxian Hospital, Southern Medical University, Shanghai, 201400, China. [5] Breast and Thyroid Surgery Center, Union Hospital of Tongji Medical College, Huazhong University of Science and Technology, Wuhan, 430022, China. [6] Department of Oncology, Guangzhou Chest Hospital, Guangzhou 510095, China. [7] Department of Pathology, Xiangya Hospital and School of Basic Medical Sciences, Central South University, Changsha, 410078, China. [8] Department of Pathology and Immunology, Baylor College of Medicine, and Michael E. DeBakey VAMC, Houston, TX 77030, USA. [9] Department of Urologic Sciences and Vancouver Prostate Centre, University of British Columbia, Vancouver, BC, V5Z 1M9, Canada. [10] Department of Pharmacy, Changzheng Hospital, Shanghai 200003, China. [11] Department of Medical Oncology, Nanfang Hospital, Southern Medical University, Guangzhou, 510515, China. [12] Department of Gynaecology and Obstetrics, Union Hospital of Tongji Medical College, Huazhong University of Science and Technology, Wuhan, 430022, China. [13] Division of Oncology, Department of Internal Medicine, and Memorial Herman Cancer Center, University of Texas Health Science Center at Houston, Houston, TX 77030, USA. [14] University of Texas MD Anderson Cancer Center UTHealth Graduate School of Biomedical Sciences, Houston, TX 77030, USA. These authors contributed equally: Yan Zhang, Dayong Zheng, Ting Zhou.  Correspondence and requests for materials should be addressed to W.L. (email: wenliang.li@uth.tmc.edu)

Androgen-deprivation therapies (ADT) are the mainstay treatment for prostate cancers. ADT is effective initially but a majority of tumors relapse with castration-resistant prostate cancer (CRPC), from which most patients eventually die. CRPC is driven primarily by aberrant activation of AR in the milieu of castrate serum levels of androgen[1]. On the other hand, approximately 25% of the men who die of prostate cancer have tumors with a neuroendocrine phenotype associated with low AR signaling and poor prognosis[2,3]. With the recent introduction of new generation potent AR pathway inhibitors, such as abiraterone and enzalutamide, the incidence of NEPC has increased, which is associated with a poor outcome[4,5]. Our knowledge of NEPC biology is still very limited and currently there is no effective treatment for NEPC. The mechanisms of CRPC progression, particularly pathways involved in the development of neuroendocrine prostate cancer (NEPC), need to be better understood for the development of future effective treatments for NEPC[3,4,6].

We and others have previously shown that ADT leads to activation of CREB, which in turn promotes neuroendocrine differentiation (NED) of prostate cancer cells[7,8]. In AR-positive prostate cancer cells, CREB-binding protein (CBP), a histone acetyltransferase, has been shown to act as an AR coactivator in transcriptional activation of AR target genes[9]. However, it is still largely unclear how CREB activation promotes AR-indifferent NEPC. Elucidation of this mechanism is crucial for our understanding and developing treatments of CRPC/NEPC.

Another mediator potentially important for NEPC is polycomb repressive complex 2 (PRC2), which establishes transcriptional repression by tri-methylating lysine 27 of histone H3 (H3K27me3)[10,11]. The major enzyme for catalyzing this histone mark is EZH2 (enhanced zeste homolog 2)[12], which is overexpressed in several solid tumors[11,13]. EZH2 expression and its PRC2 activity are particularly high in NEPC[6,14,15]. Overexpression of EZH2 in prostate cancer cells is known to promote prostate cancer cell proliferation and migration (review[11]). It remains incompletely understood whether and how EZH2 directly contributes to NED, and what biological processes are responsible for elevated PRC2 activity in NEPC cells[6,16].

Angiogenesis plays a crucial role in prostate cancer survival, progression, and metastasis[17]. NEPC is known to be highly vascularized[18,19]. Angiogenesis is a complicated process that is dependent on switching the balance between activators and inhibitors of angiogenesis[20]. VEGF and several neurosecretory peptides, such as bombesin and gastrin, are known to promote angiogenesis in NEPC[21]. However, it is unknown what endogenous angiogenic inhibitors are involved in angiogenesis regulation in NEPC and whether EZH2 overexpression in NEPC cells contributes to angiogenesis. Thrombospondin 1 (TSP1 or THBS1) was the first identified endogenous inhibitor of angiogenesis. It potently inhibits angiogenesis directly by interfering with endothelial cell migration and survival, and its suppression results in increased angiogenesis[22]. Interestingly, TSP1 is among a list of potential EZH2-repressed targets in gene expression profiles of prostate cancer cells upon EZH2 modulation[23]. However, confirmation and characterization of an EZH2-TSP1 relationship was still lacking.

Molecular links between NED and angiogenesis in NEPC have been largely unclear. In this study, we have uncovered functional connections among ADT, CREB activation, EZH2-mediated epigenetic repression, NE phenotypes, TSP1 expression, and angiogenesis in prostate cancer cells. Our results indicated that ADT-activated CREB promotes angiogenesis and NED through enhancing PRC2 activity of EZH2 that in turn upregulates NE markers and downregulates TSP1.

## Results

**ADT-induced CREB activation is critical for neuroendocrine phenotype**. To determine whether androgen deprivation therapy (ADT) activates CREB, we found that enzalutamide (MDV3100) treatment leads to enhanced CREB activation (as indicated by p-CREB-S133 level) in AR-positive LNCaP and VCaP cells, which is reversed by androgen DHT (dihydrotestosterone) (Fig. 1a). In line with the notion that ADT induces NEPC progression, CREB is upregulated and activated in NEPC NCI-H660, NE1.3, and 144-13 cells, as compared to androgen-dependent prostate cancer (ADPC) LNCaP cells (Fig. 1b). NCI-H660 was isolated from a small cell prostate cancer, a tumor that is composed of pure malignant NEPC cells (ATCC). NE1.3 was derived from LNCaP cells upon long term culturing in charcoal stripped serum (CSS) medium that deprives hormone and mimics ADT[24]. 144-13 cells were derived from NEPC patient-derived xenograft (PDX) MDA PCA-144-13[25].

As expected from our previous study[7], CREB overexpression and activation induces the expression of NE markers in prostate cancer cells (Fig. 1c). Here, we further demonstrated that inhibitors of beta adrenergic and PKA/CREB signaling, such as ICI-118,551 (ICI), propranolol (PRO), and PKA inhibitor (PKI), downregulate NE markers CHGA and CHGB in NEPC NE1.3 and 144-13 cells (Fig. 1d). We next determined whether CREB itself is essential for ADT-induced NE marker expression. LNCaP cells expressing doxycycline (Dox)-inducible CREB shRNA (shCREB) or a dominate negative CREB polypeptide (ACREB)[26] were treated with CSS medium or MDV3100, respectively, without or with Dox induction. As shown in Fig. 1e, Dox-induced shCREB reduces CREB and p-CREB levels in LNCaP cells, which reverses NE marker induction by CSS. Similar effects were seen in Dox-induced ACREB with 24 h or 72 h of MDV3100 treatment in LNCaP cells (Fig. 1f). Together, these results indicate that CREB activation is induced by ADT, which is critical for ADT-induced NED of prostate cancer cells.

**EZH2 is activated in NEPC**. Consistent with the literature[27,28], treating LNCaP cells with MDV3100 or culturing another AR-positive prostate cancer cell line 22Rv1 in CSS medium has resulted in induction of H3K27me3 and NE markers (Fig. 2a), which suggests that ADT activates EZH2's PRC2 activity. EZH2 activity is known to be elevated in NEPC patient samples and genetically engineered mouse (GEM) models[6,14]. As expected, we found that H3K27me3 levels are higher in NEPC NE1.3, NCI-H660, and 144-13 cells than in ADPC LNCaP cells, as well as higher in NEPC patient-derived xenograft (PDX) MDA-PCA-144-13 tumor than in adenocarcinoma PDX MDA PCA-133 tumor[29,30] (Fig. 2b). To confirm an increase in EZH2's function in NEPC cells, we performed RT–qPCR analysis for several known EZH2-repressed targets, including SLIT2, ADRB2, and DAB2IP[23,31–33]. Indeed, their expression is lower in NEPC NE1.3 and NCI-H660 cells than in LNCaP cells (Fig. 2c). Their expression is also lower in NEPC PDX MDA PCA-144-13 tumor than in adenocarcinoma PDX tumor MDA PCA-133 tumor (Fig. 2d). These results confirm that EZH2 activity is higher in NEPC cells and elevated by ADT.

**ADT activates EZH2 through PKA/CREB signaling**. Our preceding results are consistent with the literature that has separately indicated an elevation of CREB and EZH2 activity in NEPC cells. However, these two proteins have not been directly linked in NEPC and prostate cancer progression. We hypothesized that ADT and CREB activation enhances EZH2 epigenetic activity. To test this hypothesis, we employed pharmacological and genetic perturbations for CREB activity, followed by examination of

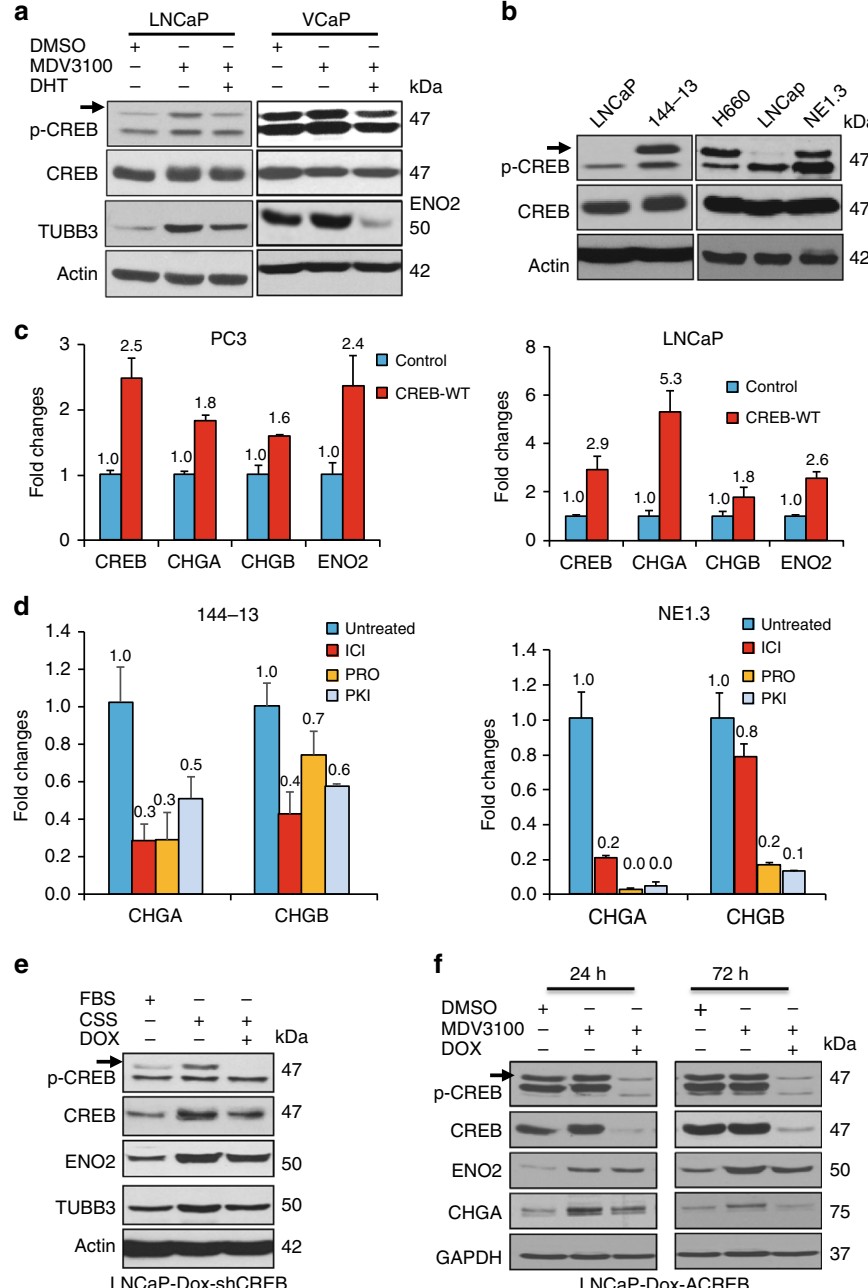

**Fig. 1** Induced by ADT, CREB activation is critical for neuroendocrine phenotype of NEPC cells. **a** Western blotting for p-CREB (S133, activated) and NE markers ENO2 and TUBB3 in AR-positive LNCaP and VCaP cells treated with 10 μM ADT drug MDV3100 (enzalutamide) for 72 h, without or with androgen 5 nM DHT during the last 24 h before cell lysate was collected. The upper band in p-CREB western blots is pS133-CREB and the lower band is p-ATF1. **b** Western blotting for CREB activation level (p-CREB) in NEPC NE1.3, NCI-H660, and 144-13 cells, as compared to LNCaP cells. **c** RT-qPCR for NE markers CHGA, CHGB, and ENO2 in prostate cancer cells overexpressing CREB wild-type (WT) cDNA. Y-axis shows relative fold changes in expression, normalized to GAPDH. **d** RT-qPCR for NE markers CHGA and CHGB in NEPC 144-13 and NE1.3 cells treated with Protein Kinase A inhibitor PKI (10 μM), beta adrenergic antagonists ICI (10 μM) or propranolol (PRO, 10 μM) for 48 h. Error bars in PCR results represent standard deviation (s.d.) of triplicate experiments. **e** LNCaP cells carrying Dox-inducible-shCREB were grown in regular FBS medium, or CSS medium without or with 1 μg ml$^{-1}$ of Dox for 6 days, followed by western blotting. **f** LNCaP cells carrying Dox-inducible ACREB (encoding a CREB inhibitory peptide) were treated with DMSO, 10 μM MDV3100, or 10 μM MDV3100 plus 1 μg ml$^{-1}$ of Dox for 24 and 72 h, followed by western blotting

EZH2 activity, as assessed by H3K27me3 level and EZH2 target gene expression. As showed in Fig. 3a, MDV3100 increases p-CREB and H3K27me3 levels in LNCaP cells, which is reversed by Dox-induced shCREB expression and CREB silencing. Conversely, overexpressing either wild type or constitutively active CREB cDNA induces H3K27me3 level in PC3 cells (Fig. 3b). Similarly, CREB activation by treatment with forskolin (FSK) or

isoproterenol (ISO) also leads to an increase in H3K27me3 level (Fig. 3c left panel, and Supplementary Fig. 1a). On the other hand, inhibition of CREB signaling through inhibitor treatments reduces H3K27me3 level in NEPC NE1.3 and 144-13 cells (Fig. 3c right panel, and Supplementary Fig. 1b). Accordingly, CREB signaling activator ISO or FSK represses, while CREB pathway inhibitor PRO or ICI induces, expression of EZH2-repressed

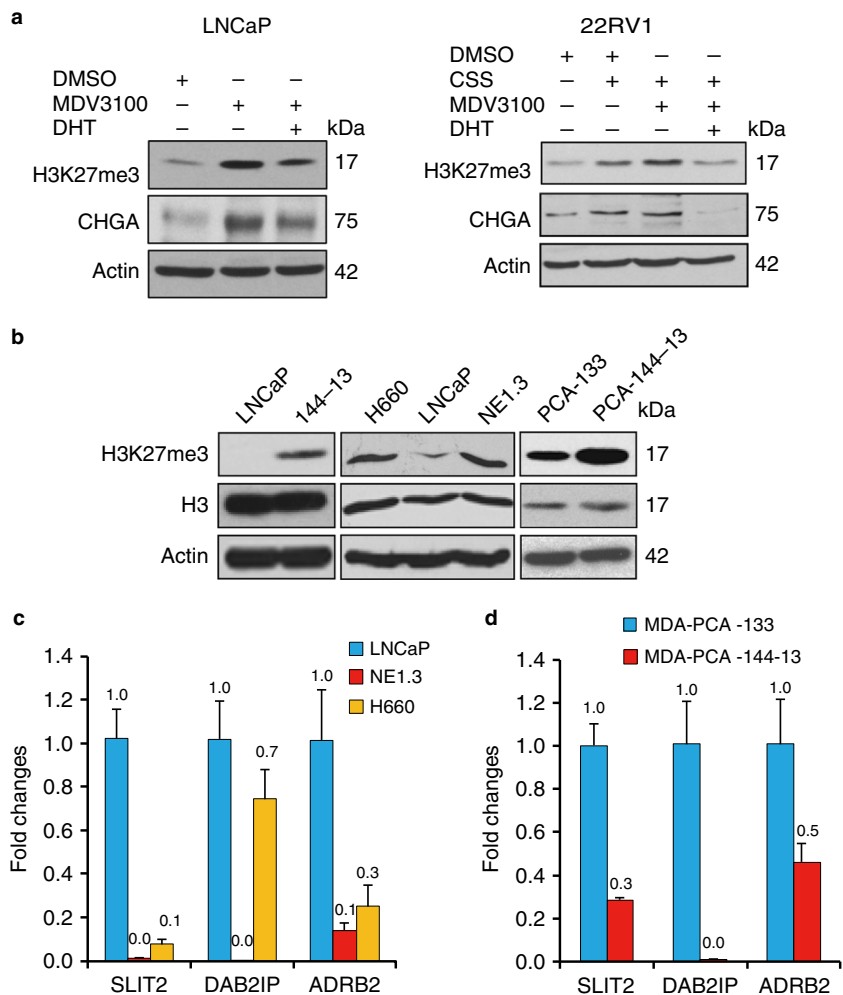

**Fig. 2** EZH2 is activated in neuroendocrine prostate cancer (NEPC). **a** Western blotting for EZH2 catalytic product H3K27me3 level and NE marker CHGA in LNCaP and 22Rv1 cells treated with either CSS or 10 μM MDV3100 for 72 h, without or with 5 nM DHT in the last 24 h. **b** Higher levels of H3K27me3 marks in NEPC NE1.3, NCI-H660, and 144-13 cells, as compared to LNCaP cells. H3K27me3 levels were also examined in patient-derived xenograft (PDX) NEPC MDA-PCA-144-13 tumor and adenocarcinoma MDA-PCA-133 tumor. Beta actin and total histone 3 (H3) were also examined as loading controls. **c**, **d** Downregulation of known EZH2 targets, SLIT2, DAB2IP, and ADRB2 in **c** NEPC NE1.3 and NCI-H660 cells as compared to LNCaP cells; and in **d** NEPC PDX tumor MDA-PCA-144-13 as compared to adenocarcinoma PDX tumor MDA-PCA-133. Y-axis shows relative fold changes in mRNA expression, normalized to GAPDH. Error bars in PCR results represent standard deviation (s.d.) of triplicate experiments

targets DAB2IP and ADRB2 (Fig. 3d, e). Moreover, the induction of H3K27me3 by ADT (via MDV3100 treatment) is abrogated by androgen (DHT) or CREB signaling antagonist PRO in LNCaP cells (Fig. 3f) and in 22Rv1 cells (Supplementary Fig. 2a). Similar results were obtained when using CSS medium as an approach for ADT in LNCaP cells (Supplementary Fig. 2b). These results indicate that ADT activates EZH2 through PKA/CREB signaling.

To further characterize the link between CREB and EZH2 activation in patient samples, we measured the levels of H3K27me3 and CREB activation (by pS133-CREB) in a tissue microarray (TM) with 78 cases of human prostate cancer and normal samples. The p-CREB level was found to positively correlate with the level of H3K27me3 ($X^2 = 16.4$, $P = 0.003$) (Fig. 3g and Supplementary Table 1). These results indicate that the CREB/EZH2 axis may be active in human prostate tissues. Overall, these results support our hypothesis that CREB activation promotes EZH2's PRC2 functions.

**EZH2 is critical for CREB induction of NE phenotypes**. We next determined whether EZH2 is critical for NE phenotypes

promoted by ADT and CREB activation. EZH2 has been shown to be overexpressed and/or activated in NEPC[6,14]. However, direct evidence of EZH2 in promoting NE phenotypes is scarce. We first showed that EZH2 indeed enhances NE marker expression. Overexpressing EZH2 increases H3K27me3 level and NE markers CHGA and CHGB in PC3 and LNCaP cells (Fig. 4a). On the other hand, inhibiting EZH2 by its inhibitor GSK126 or DZNEP decreases H3K27me3 level and NE marker expression in NEPC NE1.3 cells (Fig. 4b). Reduction of NE marker expression could also be achieved by expressing a validated EZH2 shRNA[34,35] in NE1.3 cells (Fig. 4c). Reduction of NE markers by shEZH2 was also observed in PC3 cells, which is rescued by overexpressing EZH2 cDNA (Fig. 4d). In determining whether EZH2 is critical for CREB-induced NE phenotypes, we found that CREB signaling activator FSK could no longer induce NE markers when the cells were co-treated with EZH2 inhibitor GSK126 or shEZH2 (Fig. 4e, f and Supplementary Fig. 2c). As expected, GSK126 alone reduces NE marker CHGA (Supplementary Fig. 2c).Together, these results indicate that EZH2 plays a critical role in promoting NE phenotypes and in mediating the process of CREB activation-mediated NE transition.

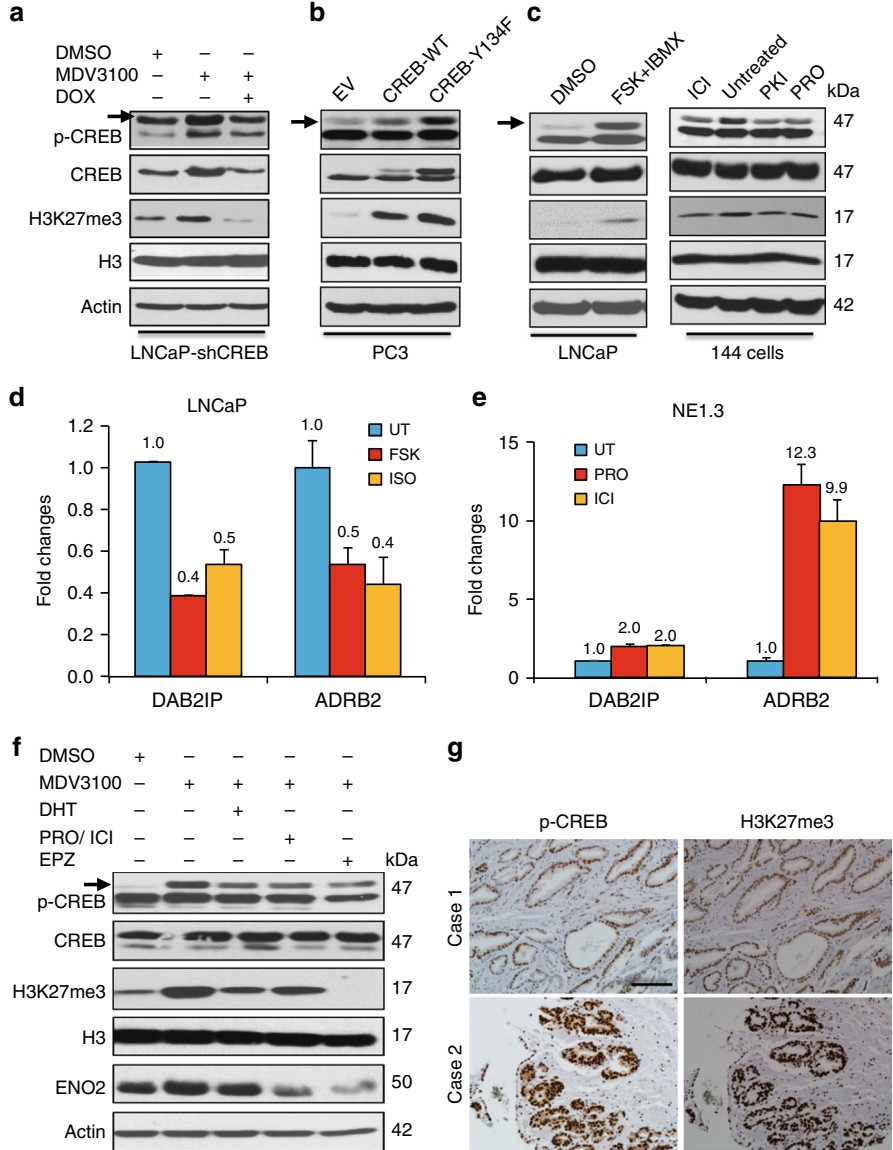

**Fig. 3** ADT activates EZH2 through PKA/CREB signaling. **a** Western blotting for LNCaP-Dox-shCREB cells treated with 10 µM MDV3100 without or with 1 µg ml$^{-1}$ Dox for 72 h. **b** Elevated H3K27me3 levels in PC3 cells overexpressing wild-type (WT) or constitutively active mutant (Y134F) of CREB cDNA. **c** LNCaP cells were treated with PKA/CREB signaling activators, i.e., 10 µM forskolin (FSK, adenylyl cyclase activator) with 0.5 mM IBMX (phosphodiesterase inhibitor) for 24 h (left). Similar experiments for three other cell lines (PC3, NE1.3, and RWPE) treated with these PKA/CREB activators are shown in Supplementary Fig. 1a. Conversely, NEPC cells 144-13 were treated with PKA inhibitor PKI, beta-adrenergic antagonist ICI or propranolol (PRO) (all 10 µM, 48 h) (right). Similar experiment on another NEPC line NE1.3 was shown in Supplementary Fig. 1b. **d**, **e** RT-qPCR analyses of EZH2 target genes DAB2IP and ADRB2 in LNCaP cells treated with 10 µM PKA signaling activator forskolin or ISO (**d**), and in NE1.3 cells treated with PKA signaling inhibitor ICI, or PRO (**e**) (all 10 µM, 48 h). **f** Western blotting of LNCaP cells treated as indicated for p-CREB, H3K27me3, and NE marker ENO2. 10 µM MDV3100 with or without PRO + ICI (CREB signaling inhibitor, 10 µM each) or 5 µM EPZ6438 (EZH2 inhibitor) for 72 h. 5 nM of DHT was added in one of the MDV3100 treated cells 24 h before sample collection. Similar results were observed in another AR-positive cell line 22Rv1 (Supplementary Fig. 2a). **g** A tissue microarray with 78 cases of human prostate cancer and normal samples was IHC stained with antibodies against pS133-CREB and H3K27me3. IHC pictures for two representative cases are shown (scale bar = 50 µm). The summary of IHC data is in Supplementary Table 1

**TSP1 is a novel EZH2 target in prostate cancer cells**. Previously we reported that CREB activation promote angiogenesis in part through reducing potent anti-angiogenic protein TSP1 expression in cancer cells[36]. NEPC is known to be highly angiogenic[21]. So far we have shown that the CREB/EZH2 axis is important for neuroendocrine phenotypes of prostate cancer cells. We next tested whether the effects of CREB activation in promoting NE phenotypes and angiogenesis are connected through EZH2. EZH2 is induced by VEGF in endothelial cells, which contributes to angiogenesis[37]. It was unclear whether EZH2 expression in cancer cells is critical for angiogenesis. Interestingly, TSP1 was among a list of genes identified as potential EZH2 repressed targets in prostate cancer cells, with no further validation[23]. We postulated that NE phenotype and angiogenesis are connected in NEPC cells through the CREB/EZH2/TSP1 pathway.

We first examined whether TSP1 is indeed an EZH2 target in prostate cancer cells. In several prostate cancer cell lines, overexpressing EZH2 represses TSP1, while suppressing EZH2 with inhibitors or shRNA induces TSP1 (Fig. 5a–c). Chromatin Immunoprecipitation (ChIP) with H3K27me3 Ab, followed by

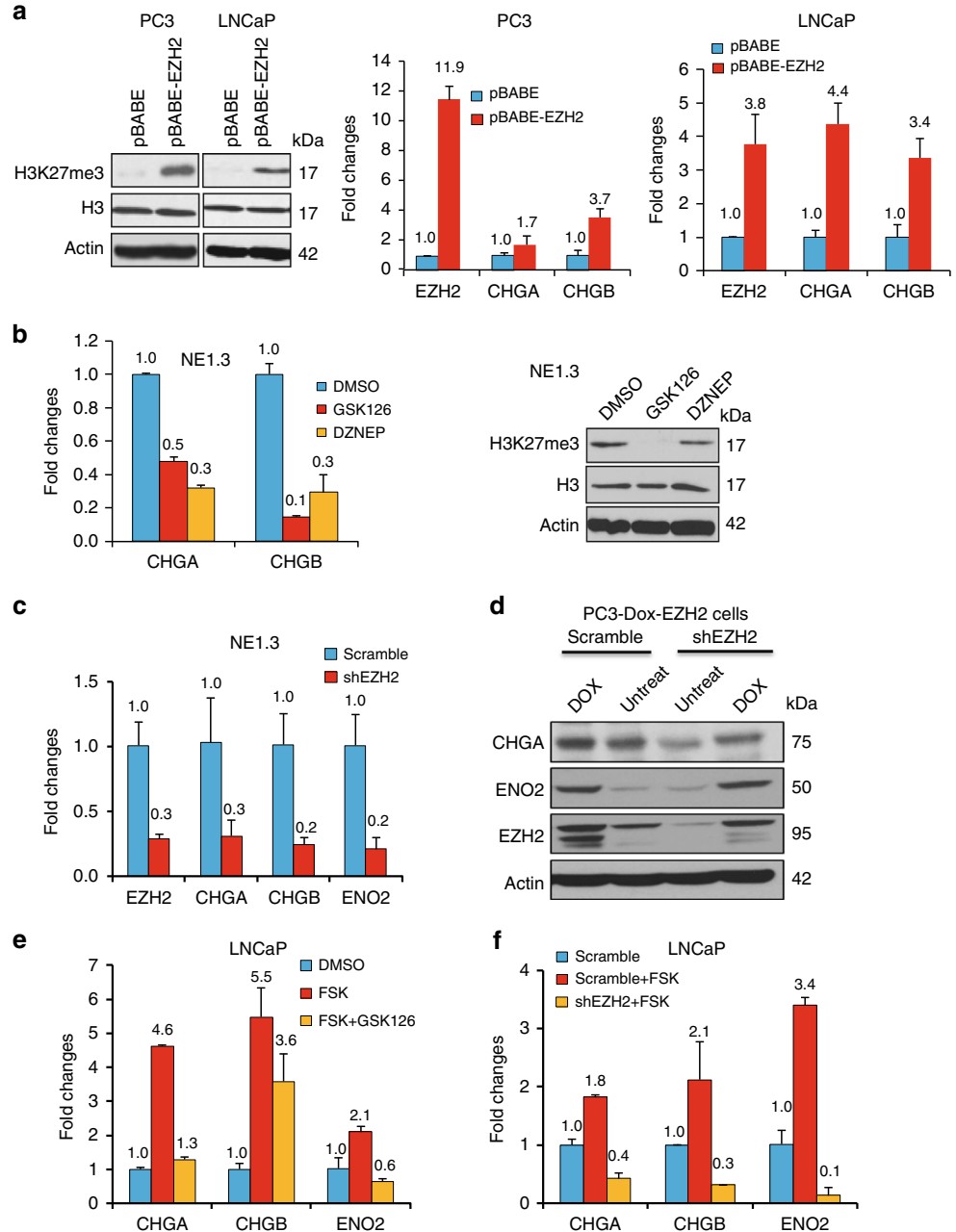

**Fig. 4** EZH2 is a critical for CREB to induce NE phenotypes. **a** H3K27me3 level (left), as well as expression of EZH2 and NE markers (right), are up-regulated in PC3 and LNCaP cells overexpressing EZH2 cDNA. **b** RT-qPCR measures the expression of NE markers CHGA, CHGB in NE1.3 cells treated with EZH2 inhibitors GSK126 (5 μM) or DZNEP (5 μM) for 48 h (left). Western blotting confirmed the reduction of H3K27me3 in NE1.3 cells treated with these EZH2 inhibitors (right). Error bars in PCR results represent standard deviation (s.d.) of triplicate experiments. **c** RT-qPCR result shows that EZH2 knockdown by a validated shRNA downregulates the expression of NE markers CHGA and CHGB in NE1.3 cells. **d** PC3 cells carrying Dox-inducible EZH2 cDNA were infected with scramble shRNA or shEZH2 lentiviruses, then treated without or with 1 μg ml$^{-1}$ of Dox for 72 h. **e** RT-qPCR analysis of the changes in NE marker expression in LNCaP cells after treating with 10 μM forskolin (FSK) with or without 5 μM EZH2 inhibitor GSK126 for 24 h. **f** LNCaP cells expressing either control shRNA or shEZH2 were treated with 10 μM FSK for 5 h. CHGA, CHGB, and ENO2 levels were measured by RT-qPCR, using GAPDH as the loading control

PCR of TSP1 promoter sequence revealed that TSP1 promoter is occupied with H3K27me3 mark, which is reduced by EZH2 inhibitor DZNEP (Fig. 5d). Using a TSP1 promoter luciferase construct[38], we further demonstrated that transcriptional activity of TSP1 promoter is increased upon treatment of EZH2 inhibitor DZNEP (Fig. 5e). Moreover, upregulation of TSP1 by siEZH2 can be rescued by expressing a siRNA-resistant EZH2 cDNA (Fig. 5f). Collectively, these data establish that TSP1 is an EZH2 repressed target in prostate cancer cells.

**TSP1 expression negatively correlates with EZH2 and NE markers.** Further supporting the existing of an EZH2/TSP1 axis in human prostate cancers, we found that TSP1 and EZH2 are expressed significantly lower and higher, respectively, in meta-static CRPC than in localized prostate cancers (Grass-o_mCRPC[39]) (Fig. 6a). Similarly, in the Beltran_NEPC dataset[6], TSP1 expression is lower, while EZH2 expression is higher, in NEPC than in CRPC-adenocarcinoma (Fig. 6b). As expected, NE markers are upregulated in metastatic tumors vs localized tumors,

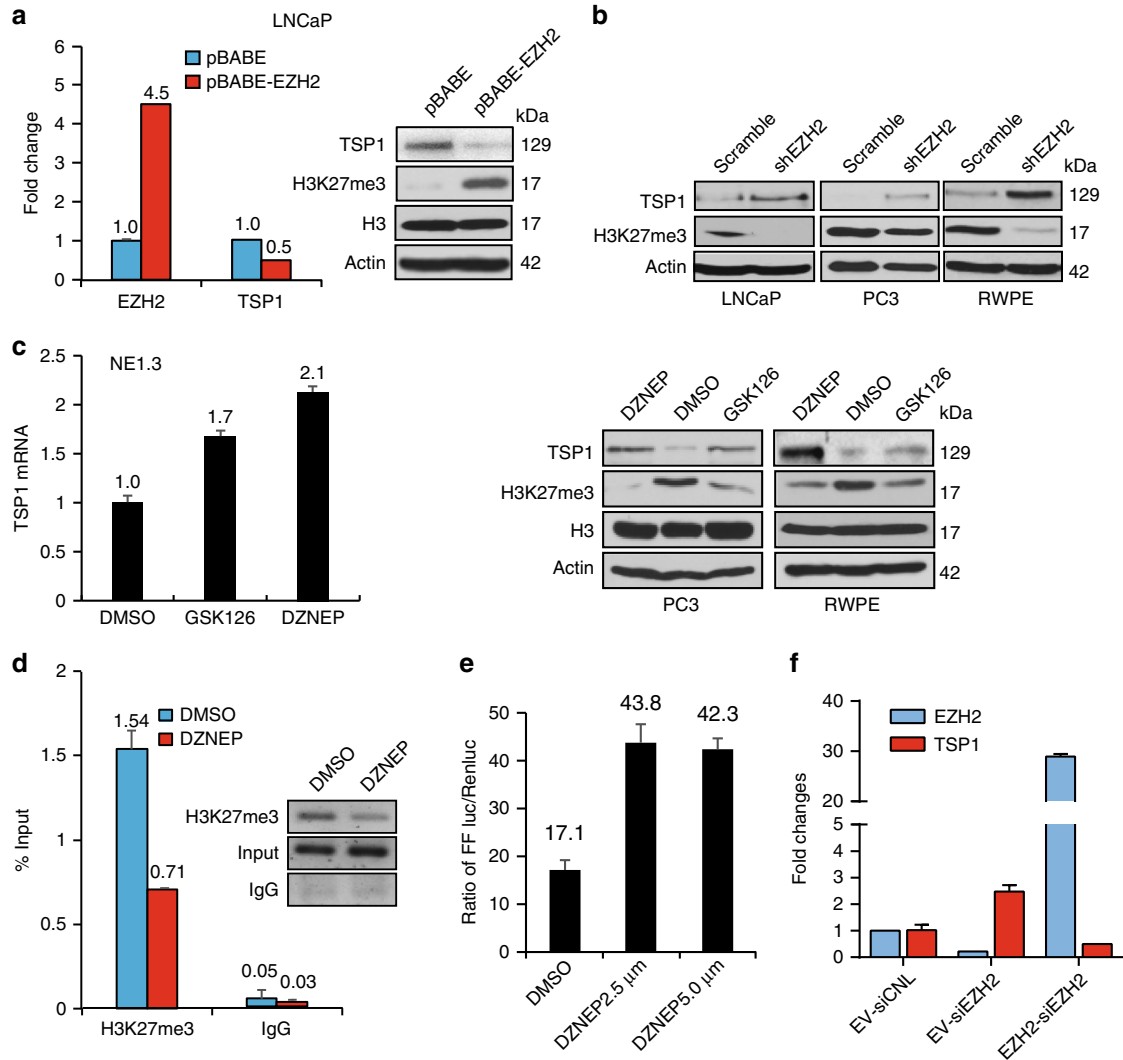

**Fig. 5** TSP1 is a novel EZH2 epigenetic target in prostate cancer cells. **a**, **b** Changes in TSP1 expression and H3K27me3 levels upon EZH2 cDNA overexpression (**a**) and shRNA silencing (**b**) in prostate cancer cells. **c** Marked reduction of H3K27me3 and up-regulation of TSP1 in PC3, RWPE, and NE1.3 cells treated with 5 μM of EZH2 inhibitor DZNEP or GSK126 for 24 h. **d** PC3 cells were treated with 2.5 μM DZNEP or vehicle control (DMSO) for 24 h. A ChIP assay was conducted using IgG control or anti-H3K27me3 antibody. The level of H3K27me3 mark on TSP1 promoter was presented as % of the inputs (*Y*-axis), which was confirmed through DNA gel electrophoresis. **e** PC3 cells were transfected with pGL3 firefly luciferase vector with TSP1 promoter along with a pRL-TK renilla luciferase control vector, then treated with DZNEP 2.5 μM and 5 μM for 24 h, followed by Dual-luciferase reporter assay. Plotted on *Y*-axis are the ratios of firefly luciferase signals vs renilla luciferase control signals in each treatment. **f** PC3 cells carrying an empty vector or siRNA-resistant EZH2 cDNA were transfected with control siRNA (siCNL) or siEZH2. Total RNA was extracted 72 h after siRNA treatment, and RT-qPCR for EZH2 and TSP1 was carried out, using GAPDH as the internal control. Plotted on *Y*-axis are average fold changes with s.d.

and are also expressed higher in NEPC than in CRPC-adenocarcinoma (Fig. 6a, b, and Supplementary Fig. 3a,3b). Consistently, mRNA expressions of EZH2 and TSP1 negatively correlate with each other in human prostate cancers, such as in Grasso_mCRPC (Spearman Correlation Rho = −0.55, $P \leq$ 1E-6) and in Beltran_NEPC (Rho = −0.53, $P \leq$ 8E-5) (Fig. 6c). Furthermore, expression of NE markers CHGA and ENO2 is negatively and positively correlated with that of TSP1 and EZH2, respectively (Fig. 6d and Supplementary Fig. 3c). Similar correlations between NE markers, TSP1 and EZH2 are observed in other prostate cancer datasets, such as TCGA, SU2C[40], FHCRC[41] prostate cancer datasets (Fig. 6d and Supplementary Fig. 4). Interestingly, TSP1 expression is negatively correlated with that of EZH2 in the CCLE dataset for 1000 human cancer cell lines of many cancer types[42] (Fig. 6e, Spearman Correlation Rho = −0.51, $P \leq$ 9.96E-57), which suggests that the EZH2/TSP1 axis exist broadly in human cancer cells. Supporting this broader

implication, EZH2 and TSP1 expression negatively correlates with each other in several large TCGA solid tumor datasets, such as breast, stomach, and colorectal cancers, as well as lung squamous carcinoma (Fig. 6e and Supplementary Fig. 3d).

Consistent with TSP1's downregulation in human mCRPCs (especially NEPCs), TSP1 protein and mRNA are downregulated by ADT (via MDV3100 treatment) in LNCaP cells, which is rescued by androgen DHT (Fig. 7a, b). As expected, TSP1 is expressed at lower levels in NEPC NE1.3 and 144-13 cells than in LNCaP cells (Fig. 7c). Mining gene expression profiles in an independent study, we also found that TSP1 is downregulated, while NE markers are upregulated, when LNCaP cells were cultured long term in CSS media in vitro[43] (Fig. 7d). Notably, we observed same expression trends in a LNCaP xenograft tumor dataset[44], where LNCaP cell-derived xenografts relapsed from treatment with second generation androgen receptor targeted therapy, enzalutamide, and became NEPC (Fig. 7e). Furthermore,

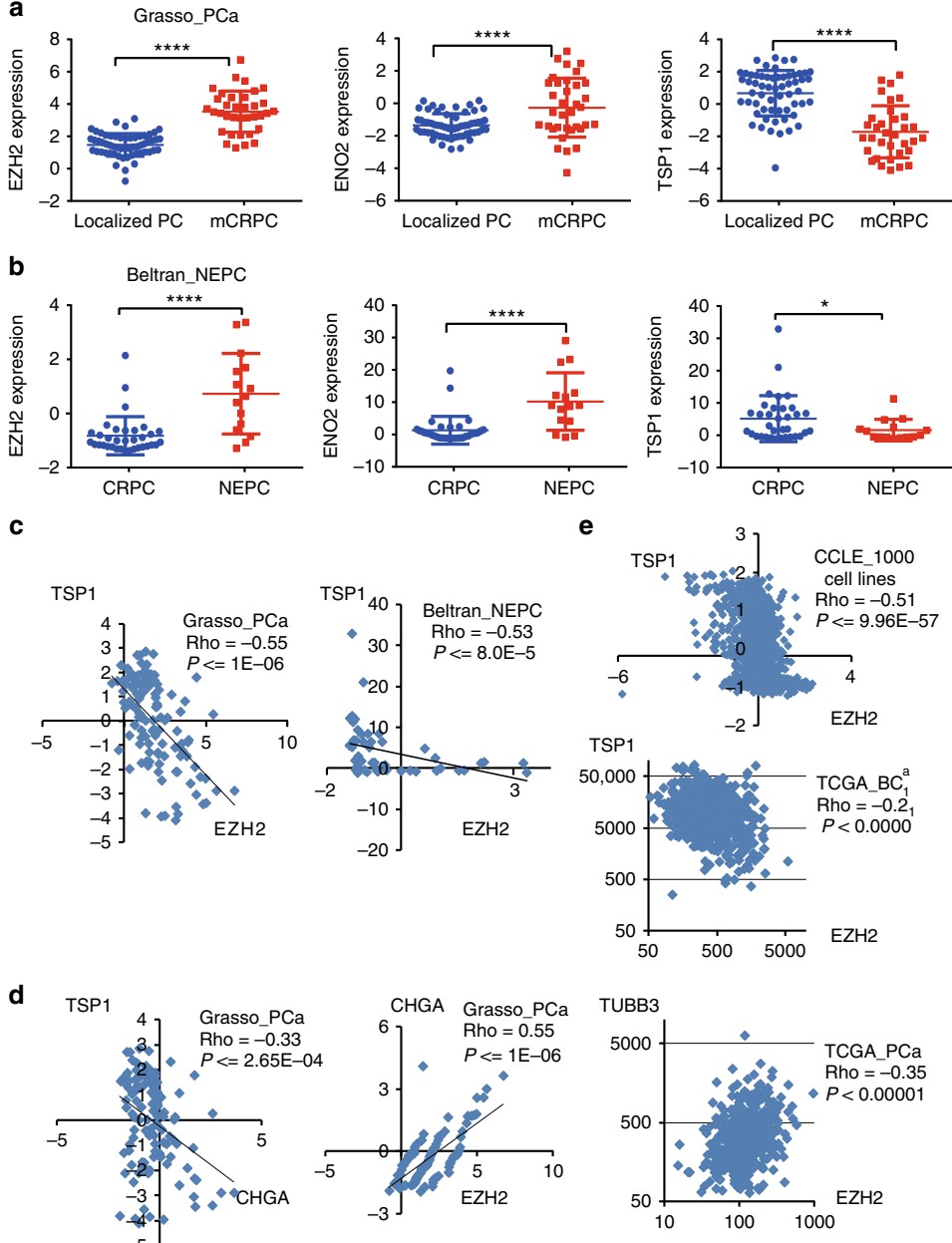

**Fig. 6** TSP1 expression is downregulated and negatively correlates with EZH2 and NE markers in NEPC. **a** Expression of EZH2 and ENO2 is higher ($P = 1.78E-11$, $P = 1.6E-3$, respectively), while TSP1 expression is lower ($P = 1.0E-9$), in metastatic CRPCs than in untreated localized prostate cancers (Grasso_PCa). **b** Expression of EZH2 and ENO2 is higher ($P = 2.95E-4$, $P = 1.4E-3$, respectively), while TSP1 expression is lower ($P = 0.017$), in NEPC than in adenocarcinoma CRPCs (Beltran_NEPC). $P$ values are from unpaired two-tailed Student's $t$ test. All values are mean ± standard deviation. ****$P \leq 0.0001$. *$P \leq 0.05$. **c** Negative correlation of TSP1 and EZH2 expression in the Grasso_PCa and Beltran_NEPC datasets. **d** The expression of NE marker CHGA is negatively and positively correlated with TSP1 and EZH2, respectively, in localized PCa and metastatic CRPC in Grasso_PCa data. In addition, EZH2 expression positively correlates with that of another NE marker TUBB3 in TCGA prostate cancer dataset. **e** Negative correlation of TSP1 and EZH2 expression in the 1000 Cancer Cell Line Encyclopedia (CCLE) dataset and TCGA breast cancer dataset. Spearman Correlation Coefficient Rho and associated $P$ values were indicated for each scatter plots. Additional results from patient samples are in Supplementary Fig. 3 and 4

when castration sensitive PDX tumor LTL331 became NEPC PDX tumor LTL331R after relapse from castration[45], we found that TSP1 is downregulated, while EZH2 and NE markers are upregulated (Fig. 7f). Additional supporting evidences came from a published dataset for prostate cancer GEM models[46], where TSP1 is expressed at significantly lower level in prostate tumors from TRAMP mice (a classic NEPC GEM model) than in prostate tumors from NP mice (a classic prostate adenocarcinoma GEM model) (GSE53202, $P = 1.36E-07$). Together, all these results

indicate that TSP1 is repressed in CRPC, especially in NEPC, and its expression is negatively correlated with those of NE markers and EZH2. They also support that the EZH2/TSP1 axis may contribute to the progression of human prostate cancers.

**ADT and CREB activation repress TSP1 through EZH2**. We next asked whether TSP1 repression by ADT and CREB activation is mediated by EZH2. We first confirmed the downregulation

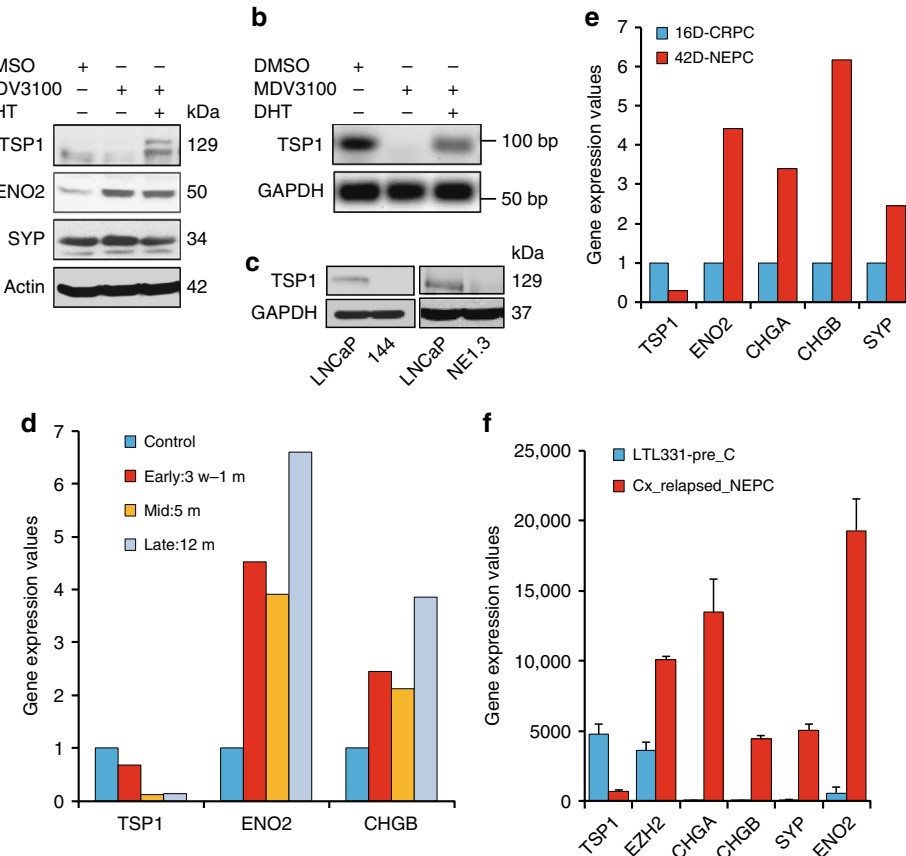

**Fig. 7** TSP1 is downregulated by ADT, and repressed in NEPC cells and xenografts models. **a** Changes of TSP1 protein expression and NE markers in AR-positive LNCaP cells treated with 10 μM MDV3100 without or with 5 nM androgen DHT for 72 h. **b** RT-qPCR of TSP1 expression in LNCaP cells treated with 10 μM MDV3100 without or with 5 nM androgen DHT for 48 h. **c** Differences in TSP1 protein expression between LNCaP cells and NEPC cells NE1.3 and 144-13. **d** Mining an independent study (GSE8702) for expression of TSP1 and NE markers after culturing LNCaP cells in CSS media for different lengths of time. **e** Expression patterns of TSP1 and NE markers obtained from transcript profiles of LNCaP xenografts[44]. 16D: xenograft resistant to castration (CRPC). 42D: xenograft resistant to castration and then became resistant to enzalutamide treatment with NE phenotypes. **f** Examining expression patterns of TSP1, EZH2, and NE markers in prostate cancer PDX transcript profiles[45] (European Nucleotide Archive access #: PRJEB9660). Prostate adenocarcinoma PDX tumors LTL331 relapse after castration and become NEPC PDX tumors LTL331R[45]

of TSP1 by MDV3100 in another AR-positive prostate cancer cell line 22Rv1 (Fig. 8a), similar to Fig. 7a for LNCaP cells. We next examined the role of CREB in repressing TSP1. Downregulation of TSP1 by CSS in LNCaP cells is reversed by PKI (Fig. 8b). Similarly, downregulation of TSP1 and induction of NE markers by MDV3100 is rescued by Dox-induced ACREB that represses CREB (Fig. 8c). On the other hand, overexpressing either wild type or constitutive active CREB cDNA represses TSP1 expression (Fig. 8d), which is rescued by simultaneous silence of EZH2 (Fig. 8e). Concordantly, CREB activation by PKA/CREB activators FSK can no longer effectively repress TSP1 expression when EZH2 is inhibited by GSK126 (Fig. 8f). Similar result was observed for another CREB signaling activator ISO when EZH2 was silenced (Fig. 8g).

To determine whether CREB activation has a direct impact on the H3K27me3 marks on the TSP1 promoter, PC3 and LNCaP cells were treated with beta adrenergic agonist ISO with or without beta adrenergic antagonist PRO, followed by ChIP with H3K27me3 Ab and PCR of TSP1 promoter sequence. As shown in Fig. 8h, the abundance of H3K27me3 marks on TSP1 promoter is increased by ISO treatment, which is reversed by additional treatment with PRO. The elevation of H3K27me3 mark on TSP1 promoter by ISO is more pronounced in LNCaP cells than in PC3 cells, which is consistent with the extent of TSP1 mRNA reduction in these two cell lines (Fig. 8i). Moreover, in line with

enhanced EZH2 activity and elevated H3K27me3 level by ADT MDV3100 as shown above, MDV3100 treatment represses TSP1, which is rescued by EZH2 inhibitor GSK126 (Fig. 8j). These results indicate that ADT and CREB activation downregulates TSP1 expression through EZH2-mediated epigenetic repression.

**CREB activation induces angiogenesis, depending on the EZH2/TSP1 axis.** Given the well-established role of TSP1 in blocking angiogenesis, we next investigated whether activation of beta-adrenergic signaling increases tumor angiogenesis through the CREB/EZH2/TSP1 pathway. Indeed, ISO treatment in prostate cancer cells induces angiogenesis tube formation of SVEC4-10 endothelial cells, which is reversed either by treating prostate cancer cells with EZH2 inhibitor GSK126 or EPZ6438, or by adding TSP1 peptide to the conditioned medium of cancer cells (Fig. 9a). We next directly assessed the contribution of EZH2 expression in cancer cells to angiogenesis. Silencing EZH2 expression by shRNA in prostate cancer cells abrogates ISO-induced tube formation of SVEC4-10 endothelial cells (Fig. 9b). Treatment of NEPC NE1.3 cells with EZH2 inhibitor GSK126 or EPZ6438 also inhibits angiogenesis tube formation of SVEC-40 endothelial cells (Fig. 9c). Moreover, we evaluated the effects of prostate cancer cells on endothelial cell migration, another in vitro assay for angiogenesis, and found that conditioned media

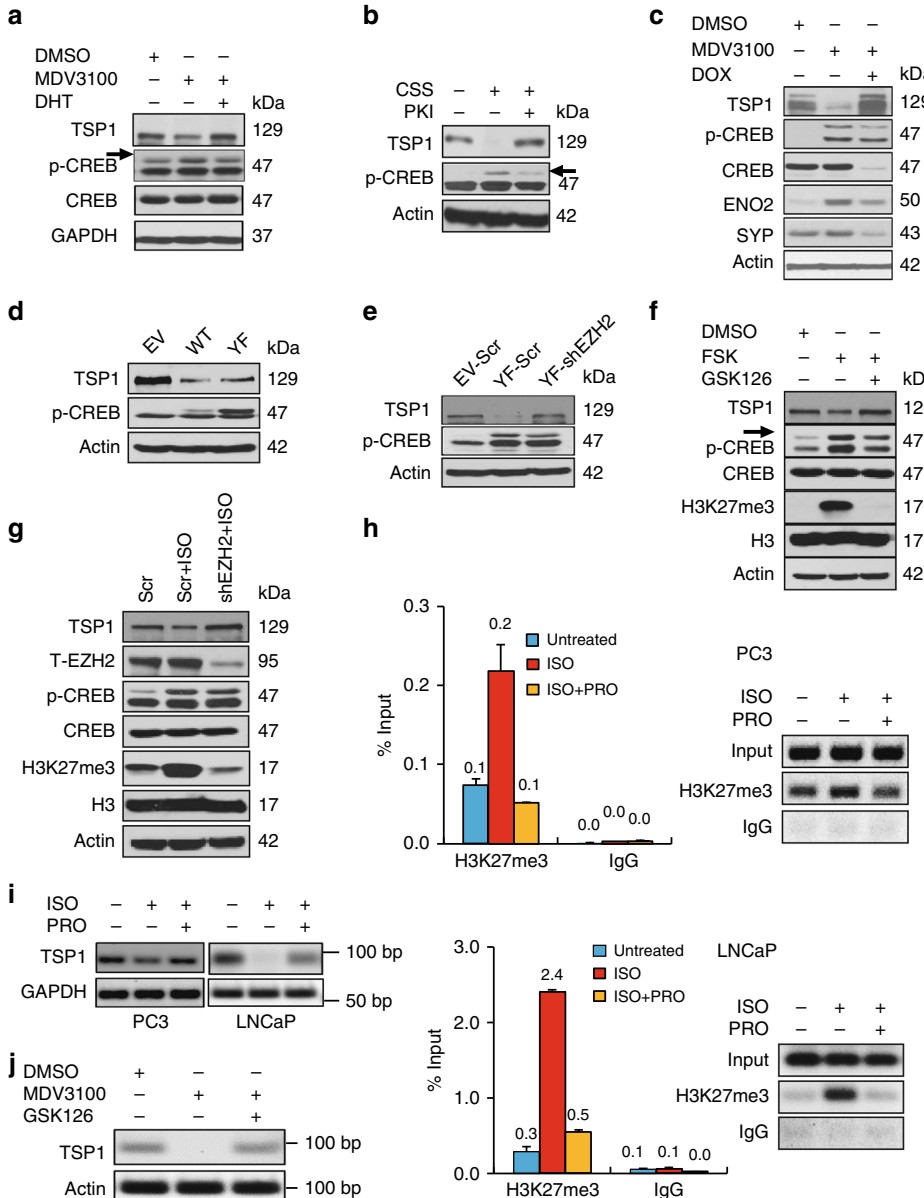

**Fig. 8** ADT and CREB activation downregulate TSP1 through EZH2-mediated epigenetic repression. **a** Western blotting for TSP1 and p-CREB levels in 22Rv1 cells treated with 10 μM MDV3100 for 72 h without or with 5 nM of DHT for 24 h. **b** LNCaP cells were grown in CSS medium for 48 h, without or with subsequent treatment of 10 μM PKA inhibitor PKI for 5 h. TSP1 and p-CREB protein levels were assessed by western blotting. **c** Protein level changes of p-CREB, TSP1, and NE markers in LNCaP-Dox-ACREB cells treated by MDV3100 with and without Dox for 72 h. ACREB, a CREB inhibitory polypeptide, is induced by Dox. **d** TSP1 expression in PC3 cells overexpressing empty vector, wild-type (WT) or constitutively active mutant (Y134F) of CREB cDNA. **e** PC3 cells expressing empty vector or CREB-Y134F were infected with pLKO.1 lentivirus for shEZH2, followed by the examination of TSP1 expression. **f** PC3 cells were treated with 10 μM PKA/CREB activator forskolin (FSK) with or without EZH2 inhibitor GSK126 (5 μM) for 24 h. Levels of indicated proteins were measured by western blotting. **g** PC3 cells expressing shEZH2 were treated with 10 μM PKA/CREB activator ISO for 24 h. PC3 cells expressing Scramble control shRNA were used as control. **h** LNCaP and PC3 cells were treated with 10 μM PKA/CREB signaling activator ISO, with or without 10 μM PKA/CREB signaling inhibitor PRO, for 24 h, followed by ChIP assay using anti-H3K27me3 or IgG control antibody. The amount of H3K27me3 histone marks on TSP1 promoter was measured by quantitative PCR and presented as % of the input (Y-axis). DNA gel electrophoresis was performed to visualize the PCR results. **i** RT-PCR for TSP1mRNA changes in PC3 and LNCaP cells that were treated similarly as in the ChIP-qPCR above. **j** LNCaP cells were treated either with DMSO, 10 μM MDV3100 without or with 5 μM GSK126 for 72 h. RT-PCR was done for TSP1 and beta actin

from NEPC NE1.3 cells induced more migration of SVEC4-10 endothelial cells than the conditioned medium from ADPC LNCaP cells (Fig. 9d). We further determined whether TSP1 induction in prostate cancer cells is critical for angiogenesis inhibition upon blockade of CREB signaling. Interestingly, effective TSP1 silencing by two independent shRNAs induces NE markers in PC3 cells (Fig. 9e). As expected, conditioned medium from PC3 expressing an effective shTSP1 attracts more

endothelial cell migration than that from PC3-scramble cells. Consistent with above results from angiogenesis tube formation assays, condition medium from PC3 cells treated with CREB signaling inhibitors PRO and ICI attracts fewer migrated SVEC4-10 cells than condition medium from untreated PC3 cells (Fig. 9f). However, PRO treatment in PC3-shTSP1 cells could no longer inhibit SVEC4-10 migration, which suggests that TSP1 is critical for angiogenesis inhibition by PRO treatment of PC3 cells

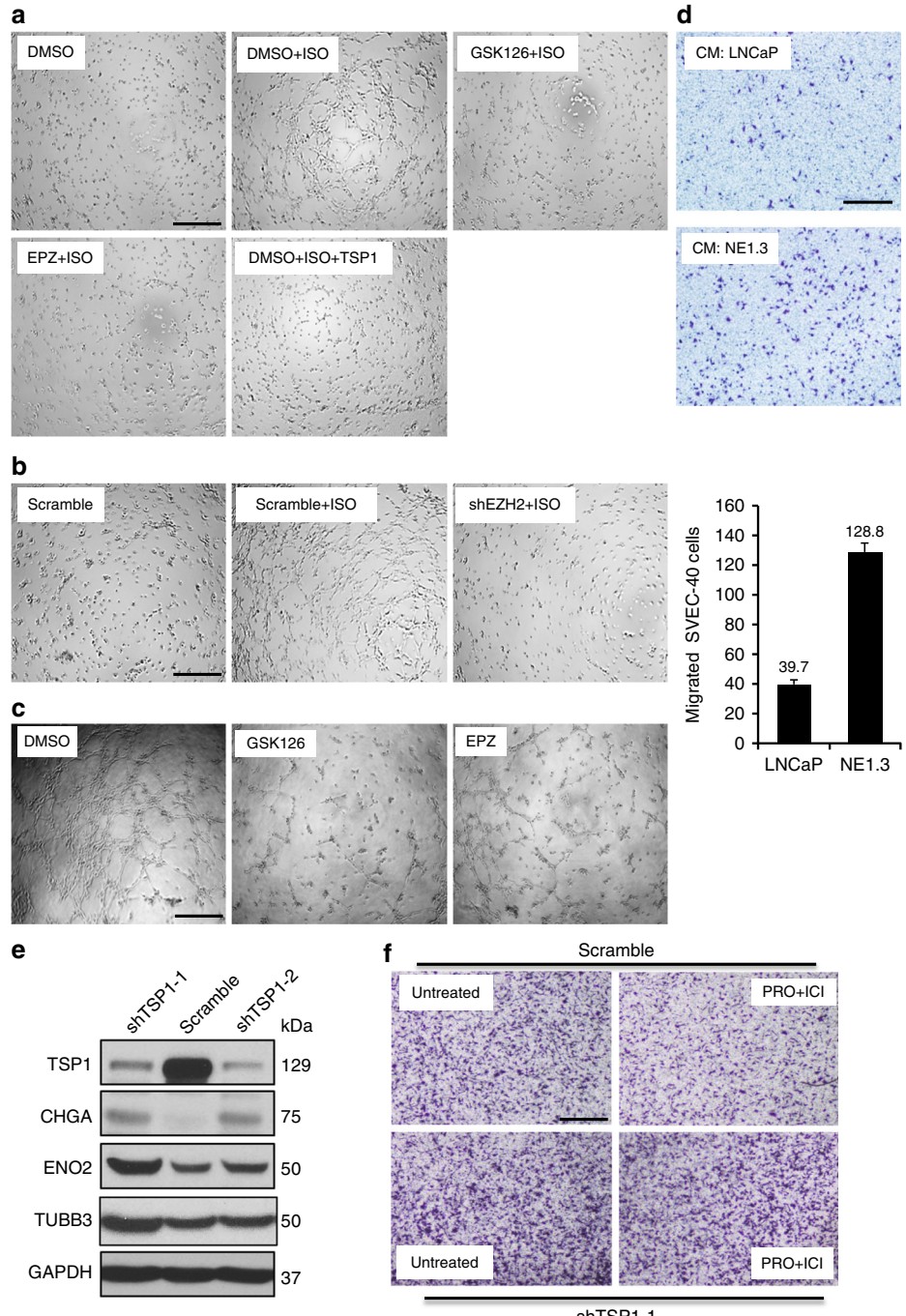

**Fig. 9** CREB activation induces angiogenesis in vitro, depending on the EZH2/TSP1 axis. **a** Serum starved PC3 cells were treated with 10 μM ISO with or without 5 μM EZH2 inhibitor GSK126 or EPZ6438. Conditioned media (CM) from the PC3 cells were then used to culture serum starved SVEC-40 endothelial cells seeded on growth factor reduced Matrigel for angiogenesis tube formation assay. TSP1 peptides were added to 10 μM of final concentration to the SVEC-40 cells in conditioned media from ISO-treated PC3 cells. The assays have been done at least twice with similar results. **b** Conditioned medium from ISO-treated PC3 cells with EZH2 knockdown can not induce angiogenesis tube formation of SVEC-40 endothelial cells. **c** Conditioned medium from NE1.3 cells treated with 5 μM EZH2 inhibitor EPZ6438 or GSK126 were used in SVEC-40 tube formation assay. **d** Conditioned medium from serum starved LNCaP and NE1.3 cells was used as attractants for migration of SVEC-40 cells in Boyden Chamber migration assay. Representative pictures from two experiments are shown. **e** Western blotting of TSP1 and NE markers in PC3 expressing scramble shRNA or two independent TSP1 shRNAs. **f** Endothelial SVEC-40 migration assay with attractants of conditioned medium from PC3-scramble or PC3-shTSP1 cells, either mock treated or treated with 10 μM PKA/CREB signaling inhibitors PRO and ICI for 48 h. Representative pictures from two experiments are shown. Scale bar = 100 μm

(Fig. 9f). Taken together, these results support our hypothesis that beta adrenergic signaling induces angiogenesis and NE phenotype through the CREB/EZH2/TSP1 pathway.

**CREB repression blocks castration-activated EZH2/TSP1/NE pathway and angiogenesis.** We have previously shown that treatment with beta adrenergic agonist, ISO, promotes tumor

growth and angiogenesis of LNCaP cell-derived xenografts (CDXs)[36]. We next compared ADPC LNCaP and NEPC NE1.3 cell growth in vivo and also determined the effects of blocking PKA/CREB signaling in controlling NE1.3 cell growth and angiogenesis. NOD/SCID mice were implanted with two million LNCaP or NE1.3 cells that were previously labelled with luciferase, which were then treated twice daily with either saline for mice with LNCaP or NE1.3 CDXs, or 2 mg kg$^{-1}$ of propranolol (PRO) for mice with NE1.3 CDXs. The tumor growth rate in these mice was monitored by bioluminescence imaging. At day 25 of CDX growth, as compared to day 1 after cell implantation, LNCaP CDX tumors grow by 4.6 folds in NOD/SCID mice, while NE1.3 CDX tumors grow by 44.3 folds ($P = 0.020$), which is consistent with the notion that NEPC cells represent more aggressive prostate cancer cells[24]. Notably, NE1.3 CDXs treated with PRO only grow by 10.6 folds (Fig. 10a and Supplementary Fig. 5a), indicating that PRO significantly blocks NEPC CDX growth ($P = 0.024$).

Next, we examined the connections among CREB activation, H3K27 trimethylation, NE phenotype, TSP1 expression, and angiogenesis in mouse xenograft tumors. Firstly, IHC staining of xenograft tumors showed that the levels of p-CREB and H3K27me3 are both increased by ISO in LNCaP CDXs and both reduced by PRO in NE1.3 CDXs (Fig. 10b), which is consistent with their positive correlation in human patient samples (Fig. 3g). Secondly, western blotting results confirm the IHC results on p-CREB and H3K27me3, and further show the expected TSP1 changes in xenograft tumors (Fig. 10c). Thirdly, we found that microvessel density (MVD), a common readout of angiogenesis in vivo, is induced by ISO treatment in LNCaP CDXs and reduced by PRO treatment in NE1.3 CDXs (Fig. 10d, Supplementary Fig. 5b and 5c). Lastly, castration of NOD/SCID mice carrying ADPC LNCaP xenografts activates the CREB/EZH2 axis, increases H3K27me3 levels, represses TSP1 expression, as well as induces NE marker CHGA and angiogenesis marker CD31. Importantly, Doxycycline-induction of ACREB (a CREB inhibiting polypeptide) in vivo inhibits these effects of castration and xenograft growth (Fig. 10e, f). Together, these results indicate that the CREB/EZH2/TSP1 pathway is responsive to ADT-induced CREB activation in mouse xenografts of prostate cancer cells, and CREB signaling is critical for castration-induced EZH2 activation, angiogenesis, and NE phenotypes in vivo.

In summary, the results in our study illuminate that: (1) CREB activation enhances EZH2's PRC2 activity; (2) ADT activates the CREB/EZH2 axis to promote NED and angiogenesis; (3) NED links to angiogenesis and tumor progression in prostate cancer cells through EZH2-mediated epigenetic repression of TSP1; (4) This pathway is activated by castration in prostate tumor xenografts, which is reversed by repression of CREB signaling; (5) The components on this pathway are accordingly expressed in cancer patient samples. Taken together, our data propose a new model of prostate cancer progression, where ADT and beta adrenergic signaling concordantly regulate PKA/CREB activation, EZH2 activity, TSP1 expression, angiogenesis, NED, and NEPC progression (Fig. 10g).

## Discussion

The transition from castration-resistant prostate adenocarcinoma (CRPC) to NEPC has emerged as an important mechanism of treatment resistance in prostate cancer. Through in vitro and in vivo studies, we investigated the still poorly understood mechanisms of NEPC progression, from which we identified a critical signaling axis, consisting of CREB/EZH2, for NED. We also studied the underlying mechanisms of heightened angiogenesis in NEPC, where we elucidated a role of the CREB/EZH2

axis in repressing anti-angiogenic factor TSP1. Our study connects a NE regulatory pathway to angiogenesis and provides critical new insights into the mechanisms of NEPC progression.

This work illuminates a previously unappreciated relationship between CREB activation and EZH2-mediated epigenetic regulation. Although these two biological processes are both critical for CRPC and NEPC development[8,26,47] and hence are expected to be coordinated, this has not been previously demonstrated. On the other hand, CREB is generally considered to function as an epigenetic activator because its activation recruits a histone acetyltransferase, CBP, to acetylate histones and activate transcription[48]. Therefore, it was previously unexpected that CREB activation would lead to an increase in epigenetic repression via EZH2. We have found that activation of CREB, through ADT or beta adrenergic-PKA signaling, robustly increases H3K27me3 levels and NE marker expression in vitro and in vivo, which can be reduced by CREB repression, EZH2 specific shRNA and inhibitors (Figs. 3, 4, and 10). Furthermore, CREB-induced NE phenotypes, TSP1 repression and angiogenesis are dependent on EZH2 (Figs. 4, 8, and 9). These results indicate that CREB and EZH2 are indeed coordinated in CPRC progression, and CREB activation enhances EZH2's epigenetic repression function. We are actively investigating the molecular mechanism of EZH2 activation by CREB signaling. One possible mechanism is through enhancing HDAC2 expression/activity by CREB activation, as we previously demonstrated in the context of chronic biobehavioral stress[36]. HDAC2 and HDAC1 are known to cooperate with EZH2 in epigenetic repression[49,50]. Supporting this potential mechanism, our preliminary study showed that TSA, a HDAC inhibitor, reverses ADT (MDV3100) induction of H3K27me3 level and NE markers, and rescues MDV3100-repressed TSP1 expression (Supplementary Fig. 2d).

Using several AR-positive ADPC cell lines, such as LNCaP, VCaP, and 22Rv1, we have demonstrated that ADT in vitro, by treatment with enzalutamide (MDV3100) or CSS medium, activates the CREB/EZH2 axis to induce NE makers and repress TSP1. These ADT effects are reversed by addition of androgen (DHT) or repression of the CREB/EZH2 axis. Notably, similar results were found in LNCaP xenograft tumors, using castration to mimic ADT and inducible CREB repression to inhibit the axis (Fig. 10e, f). These results suggest that the CREB/EZH2 axis may play an important role in mediating the effect of ADT to promote angiogenesis and NEPC progression. Several other proteins have been identified as important regulators of NEPC progression, such as RB1, TP53, EZH2, N-MYC, AURKA, SRRM4, REST, DEK, BRN2, PEG10, and SOX2[44,45,51–55]. It would be interesting to determine to what extent our pathway is linked to these known NEPC regulators.

NEPC is highly angiogenic[18,19]. It was unknown how angiogenesis and neuroendocrine phenotype are connected and what endogenous angiogenic inhibitors are involved in angiogenesis regulation in NEPC. Our work elucidates that NE phenotype and angiogenesis are connected through CREB activation of EZH2 that in turn represses anti-angiogenic factor TSP1, induces neuroendocrine markers and angiogenesis. It is still unclear how EZH2 induces NE marker expression, which is under investigation in our lab.

EZH2 has been shown to play a role in angiogenesis in endothelial cells. Lu et al.[37] reported that VEGF induces EZH2 expression in endothelial cells that in turn represses VASH1. Kottakis et al.[56] identified another pathway in endothelial cells consisting FGF2-activated NDY1-miR-101-EZH2. It was still unclear whether and how EZH2 overexpression in cancer cells contributes to angiogenesis. We have demonstrated that silencing EZH2 or treatment of EZH2 inhibitors in cancer cells inhibit in vitro angiogenesis of endothelial cells, which provides direct

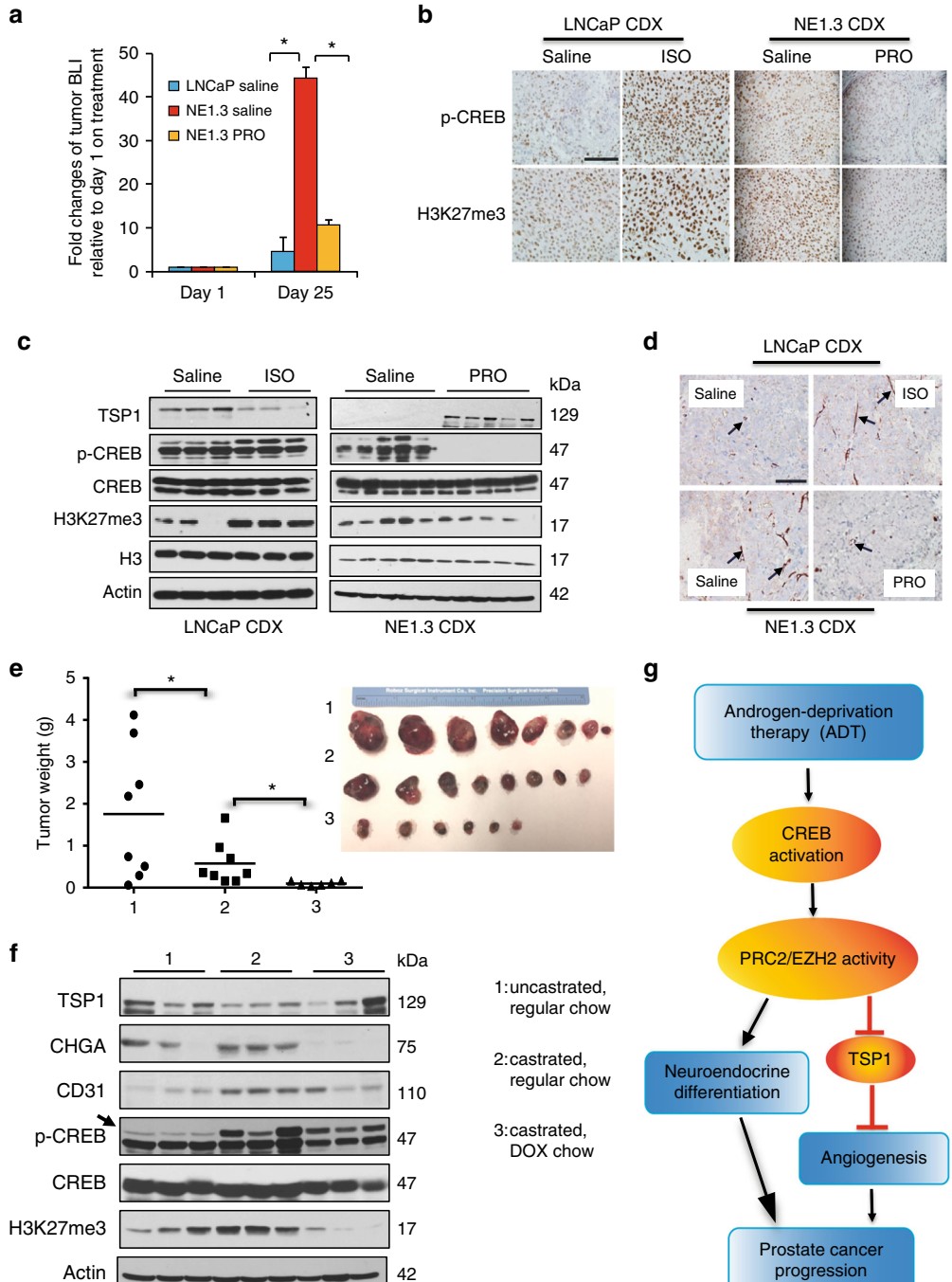

**Fig. 10** CREB repression inhibits growth of NEPC xenografts, and blocks castration-activated EZH2 axis and angiogenesis in vivo. **a** NOD/SCID mice were injected with luciferase-labelled LNCaP or NE1.3 cells, which were then treated daily with saline or 2 mg kg$^{-1}$ propranolol for 25 days, as indicated (6 mice in each group, 2 tumors in each mouse, $n = 12$). The tumor growth was monitored by bioluminescent imaging (BLI) with IVIS instrument. Presented on Y-axis are average fold changes of BLI signals for each tumor at day 25, relative to BLI signals at day 1 of treatments. **b** Left 4 panels: representative images of IHC staining for pCREB-S133 and H3K27me3 in LNCaP-derived xenografts treated daily for 25 days with saline or 10 mg kg$^{-1}$ of ISO (tumors from our previous study[36]). Right 4 panels: representative images of IHC staining for pCREB-S133 and H3K27me3 in NE1.3-derived xenografts treated daily with saline or 2 mg kg$^{-1}$ of PRO. **c** Western blots of xenograft tumors from LNCaP with or without ISO treatments (left) or from NE1.3 with or without PRO treatment (right) for the CREB/EZH2/TSP1 pathway proteins as indicated. **d** Top two panels: representative IHC staining images of angiogenesis marker CD31 in LNCaP CDX treated with saline or ISO. Bottom 2 panels: representative IHC staining images of CD31 in NE1.3 CDX treated with saline or PRO. The arrows indicated typical CD31 + microvessels. **e** Left panel: dot plots of tumor weights in the three indicated experimental groups that are collected at the end of the experiment. Right panel: photographic picture of LNCaP xenograft tumors in the three groups at sacrifice. **f** Western blotting of the proteins on the CREB/EZH2/TSP1/NE pathway in the indicated three groups of LNCaP xenograft tumors. **g** A summary model of the key findings in this study. ADT activates CREB, in part through PKA, which in turn enhances the PRC2 activity of EZH2. EZH2 is critical for ADT/CREB-induced neuroendocrine differentiation and angiogenesis of prostate cancer cells. EZH2 enhances angiogenesis through epigenetically repressing anti-angiogenic factor TSP1. Additional data related to this Figure are in Supplementary Fig. 5. Statistical significance was determined by using unpaired two-sided Student's *t* test and shown as mean with s.d. *$P < 0.05$. Scale bar $= 100$ μm

evidences supporting a role of EZH2 in cancer cells contributing to angiogenesis. We have further found that anti-angiogenic protein TSP1 is downregulated in NEPC as a target of EZH2 epigenetic repression. Intriguingly, we also found that silencing TSP1 induces NE marker expression in PC3 cells (Fig. 9e), which further supports an intimate connection between angiogenesis and neuroendocrine phenotype.

No effective treatment is currently available for NEPC. Consistent with others' reports[6,14], our study indicates a critical role of EZH2 in NEPC, which suggests that EZH2 may be a valuable target for NEPC. EZH2 inhibitor EPZ6438 is currently under several clinical trials for multiple types of lymphoma, solid tumors and synovial sarcoma, such as clinical trials NCT03010982 and NCT01897571. It is conceivable that EPZ6438 or other EZH2 inhibitors may have some efficacy in treating NEPC, which should be formally investigated in additional preclinical and clinical studies.

Notably, we found that treatment with beta adrenergic antagonist propranolol (a beta blocker) significantly reduced the growth of NEPC cell-derived xenografts in NOD/SCID mice. This result is consistent with a critical role of CREB signaling in NE phenotype and angiogenesis, as demonstrated by us and others[7,8,36,47]. Epidemiology studies have demonstrated that usage of beta blockers, especially PRO, for cardiovascular diseases in some cancer patients are associated with better clinical outcomes in multiple cancer types, such as melanoma, prostate, lung, and breast cancers[57–59]. These retrospective studies in cancer patients are in line with mounting evidences supporting an antitumor effect of PRO in cancer cell culture and in mouse xenografts[60–62]. Given that beta blockers have been safely used for decades for hypertension and heart diseases, our result suggests that beta blockers may become new therapies for NEPC, likely through combinations with other cancer therapeutics. According to clinicaltrials.gov, beta blocker PRO is being tested in several clinical trials for solid tumors. A main obstacle is the lack of clear understandings of PRO's mechanism of action in vivo as well as a shortage of biomarkers for patient selection and efficacy monitoring[58,63]. Our study suggests that PRO may exert its activity, at least in part, through regulating the CREB/EZH2/TSP1 pathway. The activity of this pathway may also become biomarkers for patient selection and efficacy monitoring, which warrants further studies in the future.

Lineage plasticity promotes conversion of a cancer cell that is dependent on a therapeutic target to another cell that is indifferent to the function of that target. Similar to conversion from AR-dependent adenocarcinomas to AR-indifferent NEPCs, some EGFR-mutant and initially EGFR-dependent lung adenocarcinomas relapse with the appearance of histologically distinct variants that lack EGFR and RB1 expression, but express neuroendocrine lineage markers[64]. Future studies are warranted to investigate the contributions of the CREB/EZH2/TSP1 pathway shown here in prostate cancer cells to the lineage plasticity of lung cancer cells.

## Methods

**Cell culture**. 22Rv1, RWPE, LNCaP, and NCI-H660 cells were purchased from ATCC. The PC3 prostate cancer cells used in this study represent a poorly metastatic PC3 variant that was kindly provided by Fidler[65] and was matched to PC3 cells from ATCC by DNA STR fingerprinting (Biosynthesis Inc). LNCaP, VCaP, 22Rv1, and PC3 cells were maintained in RPMI 1640 media (Mediatech), supplemented with 10% FBS (Gibco) and 1% penicillin-streptomycin. NEPC NE1.3 cells were derived from LNCaP cells after long term culturing in charcoal striped serum (CSS) medium[24] and cultured in phenol red-free RPMI 1640 medium supplemented with 5% CSS (Gibco) and 1% penicillin and streptomycin. The LNCaP and NE1.3 lines were matched to ATCC LNCaP profile by DNA STR fingerprinting (Biosynthesis Inc), confirming that the NE1.3 line was derived from LNCaP cells. Neuroendocrine small cell prostate carcinoma cells NCI-H660 and immortalized non-tumorigenic prostate epithelial cells RWPE-1 were obtained

from ATCC and cultured according to ATCC guidance. NEPC/SCPC cell line 144-13 were kindly provided by Maity and cultured as described[25]. 293T cells and mouse endothelial cells SVEC4-10, originally ordered from ATCC, were cultured in DMEM media (Mediatech), supplemented with 10% FBS and 1% penicillin-streptomycin. LNCaP cells carrying Doxycycline-inducible shCREB and ACREB were kindly provided by Hu and cultured as described[26]. Cultures were grown in a 37 °C incubator with 5% $CO_2$. All cell lines were routinely confirmed to be mycoplasma-free using the Lonza MycoAlert Detection kit (LT07-218).

**In vitro treatments with activators and inhibitors**. The activators and inhibitors used in this study were obtained from the following sources: ISO (Sigma), Forskolin (FSK, LC Laboratoy), IBMX (Adipogen), ICI (Tocris), PKI (Tocris), PRO (Tci America), GSK126 (Selleck), DZNEP (Cayman Chemical), EPZ6438 (Selleck), TSP1 peptide (Athens Research and Technology), MDV3100 (Apexbio) and Doxycycline (Enzo), TSA (Cayman). The doses and duration of their treatments were as indicated.

**cDNA/shRNA transduction and transfection**. All shRNA constructs are in pLKO.1 vector[66] and were purchased from Sigma-Aldrich (St. Louis, MO). For stable knockdown of EZH2, LNCaP, PC3, and RWPE cells were transduced with lentiviral particles of a validated EZH2 shRNA: CGGAAATCTTAAACCAAGAAT[34,35]. shTSP1-1: TATCATCTGGTA-TACCATTGC and shTSP1-2:CTCTCAAGAAATGGTGTTCTT. shScramble control, shRNAs against TSP1 or EZH2 were packaged into viral particles using 293T cells according to previously described method[66]. Briefly, 293T cells were seeded in 6-well plates at 1.5-million cells/well. Lentiviral vector carrying either scramble control, shTSP1, or shEZH2 shRNA was transfected, together with packaging plasmids VSVG and Delta 8.9, into 293T cells by TransIT-LT1, followed by centrifugation at 1100×g for 30 min. After initial medium change around 16 h post-transfection, the virus supernatant was collected 48 and 72 h after transfection, aliquoted and stored at −80 °C for subsequent experiments. Cells were infected with the lentivirus supernatant in the presence of 8 µg ml⁻¹ polybrene and subsequently selected with 1 µg ml⁻¹ of puromycin. For overexpression of EZH2, cells were infected with retrovirus for human EZH2 cDNA or pBABE-puro vector control[67], and subsequently selected with 1 µg ml⁻¹ of puromycin. EZH2 overexpression was also achieved through introduction of doxycycline-inducible EZH2 cDNA in pInducer lentiviral vector (selection with 400 µg ml⁻¹ of G418). siEZH2-resistant EZH2 cDNA construct was kindly provided by Xu[35] and the corresponding siEZH2 was ordered from Dharmacon. PC3 cells were transfected with a mammalian expression vector pcDNA3.1 (EV, empty vector), Flag-pcDNA3.1-CREB cDNA (WT, wild type), Flag-pcDNA3.1-CREB-Y134F (YF, constitutively active[68]), kindly provided by Berdeaux, using TransIT-LT1 transfection reagent (Mirus) and selection with 400 µg ml⁻¹ of G418.

**Reverse transcription and quantitative PCR (RT-qPCR)**. Total RNA was extracted from the indicated cells by using TRIzol Reagent (Life Technology). The RNA concentration and purity were measured by NanoDrop 2000 UV-Vis Spectrophotometer (Thermo Scientific). 2–3 µg of total RNA was used to generate cDNA using the iScript R Transcription Supermix (Bio-Rad). Real time qPCR was performed using SsoFast EvaGreen Supermix in CFX96 Thermal Cycler (Bio-Rad). PCR-based amplification was performed using the following primers:

EZH2 F: 5′-ccgctgaggatgtggatac-3′; EZH2 R: 5′-cagtgtgcagcccacaac-3′; ADRB2 F: 5′-ttcttgctggcacccaata-3′; ADRB2 R: 5′-gccaggacgatgagagacat-3′; SLIT2 F: 5′-cggagcagcaagctaaagaa-3′; SLIT2 R: 5′-gcgacagggacagcatct-3′; TSP1 F: 5′-gtcatacaacactcccacgc-3′; TSP1 R: 5′-ccagggcataggtagaagct-3′; CREB F: 5′-ggagcttgtaccaccggtaa-3′; CREB R: 5′-gcatctccactctgctggtt-3′; CHGA F: 5′-tacaaggagatccggaaagg-3′; CHGA R: 5′-ccatctcctcctcctcctct-3′; CHGB F: 5′-cacgccattctgagaagagc-3′; CHGB R: 5′-tctcctggctcttcaaggtg-3′; ENO2 F: 5′-ctgtggtggagcaagagaaa-3′; ENO2 R: 5′-acacccaggatggcattg-3′; GAPDH F: 5′-agccacatcgctcagacac-3′; GAPDH R: 5′-gcccaatacgaccaaatcc-3′; Beta Actin F: 5′-ccaaccgcgagaagatga-3′; Beta Actin R: 5′-ccagaggcgtacagggatag-5′. GAPDH or beta actin was used to normalize RNA input with similar results. The expression levels were calculated according to the comparative CT method (ΔΔCT).

**Western blotting analysis**. Cells were washed in ice-cold PBS and lysed in lysis buffer (30 mM Tris, 200 mM NaCl, 1.5 mM $MgCl_2$, 0.4 mM EDTA, 20% Glycerol, 1% NP-40, 1 mM DTT) with complete mini protease inhibitor cocktail and PhosSTOP phosphatase inhibitor cocktail (Roche Applied Science). Protein concentrations were determined using Pierce BCA protein assay kit (Thermo Scientific). The samples were then separated by SDS-PAGE and transferred to PVDF membrane (Bio-Rad). The membrane was blocked with 5% skimmed milk in TBST for 1 h at room temperature, followed by incubation of a primary antibody overnight at 4 °C. The dilutions and catalog numbers of primary antibodies used are listed in Supplementary Table 2. After washes, the membrane was incubated with HRP-conjugated secondary antibodies for 2 h at room temperature. The blots were then detected by Pierce ECL Western Blotting Substrate (Thermo Scientific) on Blue Basic Autoradiography Films (Bioexpress). The uncropped scans of the most important Western blots and RT-PCR DNA gel pictures are presented in Supplementary Fig. 6 and Supplementary Fig. 7.

**Chromatin Immunoprecipitation (ChIP)**. DNA binding proteins in cells were cross-linked to DNA by 1% formaldehyde for 10 min at room temperature, which was quenched with glycine. Cells were then lysed in SDS Lysis Buffer (1% SDS, 10 mM EDTA, 50 mM Tris-HCl, pH 8.1 and freshly added protease/phosphatase inhibitors) and sonicated to shear DNA to 300–500 bp fragments using Branson Low Power Ultrasonic Systems 2000 LPt/LPe sonicator (Fisher Scientific). Fifty microliters of supernatant was diluted in 450 μl dilution buffer (1% Triton X-100, 2 mM EDTA, 20 mM Tris-HCl pH 8.1, 150 mM NaCl supplemented with 0.1% NP40, protease and phosphatase inhibitors). Samples were pre-cleared with protein A/G agrose beads for 2 h. Twenty microliters of the post-cleared supernatant was kept as input. The remaining supernatants were incubated overnight with 2 μg anti-H3K27me3 (Millipore) or anti-IgG antibody, followed by 1 h incubation with protein A/G agrose beads at 4 °C. The immunoprecipitates were subjected to multiple washes for 5 min each at 4 °C in low salt buffer with 150 mM NaCl, high salt buffer with 500 mM NaCl, LiCl buffer with 250 mM LiCl and finally the TE buffer. DNA was recovered after reversion of the protein-DNA cross-links with 0.2 M NaCl and proteinase K. Subsequently, DNA was extracted with phenol-chloroform and precipitated with ethanol. Five microliters of DNA was subjected to real time PCR. Primers used to measure the enrichment of TSP1 promoter DNA sequence containing H3K27me3 marks are: F (5′-tggctgtttgcagcagtcggg-3′), and R (5′-ggatctcagcacgggcaggg-3′). The enrichment of ChIP DNA was calculated as percentage of input. The PCR products were resolved electrophoretically on a 2% agarose gel and visualized by ethidium bromide staining.

**Luciferase assay**. TSP1 promoter firefly luciferase construct that contains TSP1 promoter sequence from −2033 to +750 off the transcription starting site was generously provided by Xiao[38]. Prostate cancer cells were co-transfected with the TSP1 promoter luciferase construct and a TK-renilla luciferase construct that is the internal control to normalize for transfection efficiency. Twenty-four hours later, the cells were treated with either DMSO or EZH2 inhibitor DZNEP at 2.5 μM or 5 μM for another 24 h. The luciferase activities (firefly and renilla luciferase) were determined using the Dual-Luciferase Reporter Assay System (Promega).

**Endothelial cell tube formation assay**. PC3 and NE1.3 cells were treated as indicated overnight in RPMI1640 with 0% FBS. SVEC4-10 mouse endothelial cells were grown till 70% confluence, serum starved overnight, trypsinized and resuspended in the conditioned media collected from PC3 and NE1.3 cells. Growth Factor Reduced Matrigel (Corning) was thawed at 4 °C and 50 μl were quickly added to each well of a 96-well plate and allowed to solidify for 30 min at 37 °C. The wells were then incubated at 37 °C with SVEC4-10 cells in conditioned medium (20,000 cells per well) and monitored regularly to observe the tube formation. Pictures were taken under 4× and 10× magnification and the number of branches were quantified.

**Endothelial cell migration assay**. LNCaP and NE1.3 cells were grown till 70% confluence and starved in 0% FBS RPMI1640 medium overnight. PC3 cells expressing shScramble or shTSP1 were either untreated or treated with PRO for 48 h, followed by serum starvation overnight. SVEC4-10 cells were grown till 70% confluence, starved in 0% FBS DMEM medium overnight, trypsinized and re-suspended in 0% FBS DMEM. 50,000 SVEC4-10 cells were seeded on top of each Boyden chamber insert (8 μm, BD Biosciences). Conditioned media collected from starved prostate cancer cells (treated or not) were added to the wells underneath the inserts as an attractant. SVEC4-10 endothelial cells were allowed to migrate for 4 h and the inserts were fixed and stained with crystal violet dye to observe migrated cells. Cell migration was analyzed qualitatively by counting the numbers of migrated cells within each high power fields.

**Animal experiments**. All animal studies followed protocols approved by the Animal Welfare Committee at the University of Texas Health Science Center at Houston. We have complied with all relevant ethical regulations. LNCaP and NE1.3 cells were transfected to express luciferase. Male 6–7 week old NOD/SCID mice (Charles River Laboratories) were implanted subcutaneously with two million of LNCaP or NE1.3 cells in 100 μl 1:1 of PBS and matrigel in both sides of each mouse. The mice were then randomly divided and treated twice daily with either saline for mice with LNCaP or NE1.3 xenograft tumors (8 mice, $n = 16$ tumors in either group), or 2 mg kg$^{-1}$ propranolol (PRO) for mice with NE1.3 xenograft tumors (8 mice, $n = 16$ tumors). The tumor growth rate in these mice was monitored by bioluminescence imaging. Mice were anaesthetized using isoflurane and injected intraperitoneally with 150 mg kg$^{-1}$ luciferin (Caliper Life Sciences), and tumors were imaged using an IVIS Lumina II platform (Caliper Life Sciences) and analyzed with Live Image software (Caliper Life Sciences).

For the castration and CREB in vivo repression experiment, two million of LNCaP-Dox-inducible-ACREB cells were implanted subcutaneously in both flanks of 15 male 6–7 week old NOD/SCID mice (Envigo). When the majority of LNCaP xenograft tumors were palpable, the 15 mice were randomly divided into three groups (5 mice in each group, $n = 10$): first group uncastrated and fed with regular chow, second group castrated and then fed with regular chow, third group

castrated and then fed with chow containing 200 mg kg$^{-1}$ Doxycycline (Bio-Serv). According to power calculation and common practice in the fields, xenograft tumor's sample size of 10 is sufficient to detect statistically significant differences among different groups. When the tumors in uncastrated group reached the end point, all mice in the three groups were sacrificed. Xenograft tumors were weighted, photographed and then fixed with formalin and/or snap-frozen. All xenograft tumors were included to compare the differences in mean tumor weights. No visible tumors were excluded.

**IHC staining of human prostate tissue microarray**. TMA containing 78 cases of formalin fixed and paraffin embedded normal or cancer prostate samples were obtained through clinical protocols approved by the Institutional Review Board of Baylor College of Medicine. We have complied with all relevant ethical regulations. The TMAs were dewaxed in 60 °C oven for 2 h and deparaffinized, and rehydrated through incubating in xylene and alcohol series. Tissue sections were subjected to antigen retrieval in 10 mM sodium citrate buffer (pH 6.0) in a food steamer for 45 min. The Universal Elite ABC kit (Vector Laboratories) was used for immuno-histochemistry (IHC) staining, according to the manufacturer's instruction. After suppressing endogenous peroxidase activity, the sections were incubated in normal horse serum to prevent nonspecific immunoglobulin binding, then incubated with primary antibody overnight at 4 °C. Primary antibodies used were anti-p-CREB and anti-H3K27me3 from Cell Signaling Technology. A streptavidin-HRP detection system was used to reveal specific binding. The stained slides were scored by two investigators who reached consensus, as following: staining intensity −/+, <25% positive cells (weak, score 1); staining intensity ++, 25–50% positive cells (intermediate, score 2); and staining intensity +++, >50% positive cells (strong, score 3).

**IHC staining of xenograft tumors**. For immunohistochemical staining of xeno-graft tumors, 5-μm serial sections were deparaffinized, rehydrated and subjected to antigen retrieval with 10 mM sodium citrate (pH 6.0) for 45 min. Before staining, non-specific binding was blocked by incubation with hydrogen peroxide as per-oxidase suppressor (Thermo Scientific) and normal horse serum (Vector Labora-tories) as blocking buffers, followed by incubating with 1:100 anti-CD31 (Abcam) antibodies in antibody diluent (Biogenex) at 4 °C overnight. All sections were briefly washed with phosphate-buffered saline and incubated at room temperature with horseradish peroxidase-conjugated secondary anti-rabbit antibody. Color was then developed by incubation with a DAB substrate kit (Vector Laboratories). Nuclei were counterstained blue with hematoxylin (Sigma Aldrich) and mounted in VectaMount Permanent mounting medium (Vector Laboratories). Isotype IgG controls were used as negative controls for the staining.

**Microvessel density**. MVD of tumor tissues was assessed through immunohis-tochemical analysis of the endothelial marker CD31 and determined according to the method previously described[69]. The immunostained sections were initially screened at low magnification (×50) to identify hot spots of the neovascularization. Any yellow-brown stained endothelial cell or endothelial cell cluster clearly sepa-rate from the adjacent microvessels, tumor cells and other connective tissue ele-ments was considered a single, countable microvessel. Within the hot spot area, the average vessel count in three hot spots with a 200-fold magnification in each tumor section was considered the value of MVD. The stained slides were scored by two investigators whose consensus was reported.

**Microarray data mining**. All non-TCGA and TCGA datasets indicated genes were downloaded from cBioPortal[70] and the GEO database (http://www.ncbi.nlm.nih.gov/gds). The transformed and normalized gene expression values from these sources were used in our analysis and statistical calculation.

**Statistical analyses**. Statistical analyses were performed using GraphPad software and/or online statistics tools. $P$ values were obtained through Student $t$-test with two tails and unequal variance, unless otherwise indicated. Spearman correlation coefficient and associated $P$ values for gene expression were calculated using GraphPad or a statistics tool at http://vassarstats.net/corr_rank.html and confirmed by another online tool: http://www.socscistatistics.com/tests/spearman/default2.aspx. Chi-square test was used to assess the correlation of p-CREB and H3K27me3 IHC staining on human samples. $P$ values < 0.05 are considered significant. All error bars are defined as s.d. All central values are defined as mean.

## Data availability

The authors declare that the data supporting the findings of this study are available within the article and its supplementary information files, or are available upon rea-sonable requests to the authors.

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

## Acknowledgements

We are very grateful to Dr. Rebecca Berdeaux, Dr. Kexin Xu and Dr. Wuhan Xiao for providing us the CREB cDNA constructs, siRNA-resistant EZH2 cDNA construct, and TSP1 promoter luciferase constructs, respectively. Many thanks to Dr. Chang-Deng Hu for providing the LNCaP-inducible-shCREB and -ACREB cells, to Dr. Sankar Maity for providing prostate cancer PDX models and the 144-13 NEPC cells. Our deep appreciation also goes to Dr. Hui Wang and Dr. Xiaodong Cheng for technical assistance in CD31 IHC staining. We also would like to thank Dr. Georgina Salazar for critical reading of this paper, and to other members of the Li lab for discussions and assistance. This work was supported by a Rising STARS Award from University of Texas System, as well as awards from the Cancer Prevention and Research Institute of Texas (RP170330) and American Cancer Society (RSG-17-062-01) to W.Li. The TMA used in this study was supported by National Cancer Institute P30 Cancer Center (P30 CA125123) for Human Tissue Acquisition and Pathology Shared Resource (M.I.). We apologize to the colleagues whose work was not cited due to the constraint in reference number.

## Author Contributions

W.Li, Y.Z., D.Z., T.Z., and M.H. conceived and designed the projects. Y.Z., D.Z., T.Z., H. S., M.H., N.S., Y.L., and Z.W. performed the experiments and carried out data analysis. M.I. provided the human tissue microarray. M.I., Y.Z., and W.Li. examined the IHC staining results. L.S., M.G., H.H., W.Liao., F.X., and H.W. provided technical assistance and helpful discussion. Y.Z., T.Z., and W.Li. wrote the manuscript with inputs from all authors. W.Li. supervised the project.

## Additional information

**Competing interests:** The authors declare no competing interests.



