## [Peer Review File · Nature Communications]

Reviewers' Comments:

Reviewer #1:

Remarks to the Author:

In the manuscript titled, "Androgen deprivation promotes neuroendocrine differentiation and angiogenesis through CREB-EZH2-TSP1 pathway in prostate cancers," Zhang Y et al explore connections between CREB activation, EZH2 and thrombospondin in the setting of neuroendocrine prostate cancer models and tissues. While many of the results are interesting, the study appears to be too broad in scope at the expense of providing sufficient foundational evidence to support critical aspects of the processes that are investigated.

1. Critical experiments to causally support their proposed mechanisms, as opposed to potential confounding experimental factors, are often missing. For example, in key gain- and loss-of-function experiments (e.g. Fig 4a,c; 5a,b) carrying out rescues with wild-type and mutant proteins would be important to demonstrate that the observations are causally connected.

2. Related to the above, in Figure 1A, 2A, 7A: the results in PC3 are confusing. The increase in p-CREB, H3K27me3 and decrease in TSP1, in PC3 cells is not likely to be due to androgen deprivation since these cells do not express AR. Rather, charcoal stripped serum is depleted of many factors in addition to depletion of androgens, and it is likely that one of these other factors may be responsible for the differences in the CSS vs. FBS conditions. This then calls to question whether the results in LNCaP cells are similarly due to depletion of some other factor other than androgen receptor ligands. In these experiments, rescue with androgen (DHT, or R1881) with and without modulation of AR in the CSS cells would be critical to carry out.

Reviewer #2:

Remarks to the Author:

Previous studies have found that ADT can activate CREB, and that this can promote neuroendocrine (NE) differentiation in prostate cancer (PCa) cells. Increased EZH2 also has been linked directly to NE differentiation in PCa cells. This study provides evidence indicating that CREB activation acts by increasing EZH2 activity. EZH2 has also been shown to be pro-angiogenic, at least in part by suppressing expression of the anti-angiogenic protein TSP1. Consistent with these data, this study provides evidence that CREB acts through EZH2 to suppress TSP1. While aspects of this study are largely confirmatory of previous data (such as the link between EZH2 and NE differentiation and angiogenesis), CREB regulation of EZH2 appears to be a novel finding. However, there are no data addressing the mechanism by which CREB activates EZH2, and such mechanistic data would clearly strengthen the conclusions. Moreover, while this study and previous data indicate that ADT can activate CREB (although there are concerns about effects seen in PC3 cells, see below), more data are needed to conclude that this activation in response to ADT is indeed a major pathway driving EZH2 activation and NE differentiation. As noted below, this would include examining effects of castration in vivo, ideally in a model in addition to LNCaP. Further specific issues are detailed below.

1. In Fig. 1a should show in parallel effects on NE markers. More importantly, AR expression in PC3 cells is low or undetectable and they do not respond to androgen. Therefore, there are concerns that the effect of culturing in CSS medium is unrelated to AR.

2. Fig. 1d should address whether CREB depletion with several independent RNAi (or with rescue controls) decrease NE markers. Also, do the drugs used prevent the upregulation of NE markers after ADT?

3. There are similar concerns regarding lack of AR in PC3 cells in Fig. 2a.

4. Given concerns about whether CSS is acting just via AR, Fig. 3e should further address whether addition of a direct AR antagonist (enzalutamide) stimulates CREB and H3K27me, and whether this is blocked by CREB and EZH2 inhibitors. The results should also be generalized to at least one other AR responsive PCa cell.

5. Fig. 4e should show effects of GSK126 alone to determine whether FSK has an effect when EZH2 is blocked. This should also ideally be done at shorter times. As FSK rapidly activates CREB, the authors should establish a time course for increased H3K27me and NE markers, and look at effects of EZH2 inhibition at the earliest times an effect is seen.
6. There are concerns that as FSK and other agents used are such potent and rapid inducers of CREB activity, the effects seen may not be physiological. What is the time course for CREB stimulation, H3K27me, and induction of NE markers in LNCaP cells after treatment with enzalutamide, and what is the effect of CREB inhibition?
7. Related to the above point and figure 5, does enzalutamide increase TSP1?
8. In Fig. 6, the authors should assess for EZH2 and TSP1 correlation in the much larger TCGA and SU2C data sets.
9. In Fig. 8f, effects of ISO in PC3 seem very modest. How does this compare with effects on TSP1 mRNA?
10. It cannot be concluded from Fig. 9 that the effects seen are due to repression of TSP1. The more conclusive study would be to RNAi or CRISPR deplete TSP1 and then show that CREB activation or repression no longer affects angiogenesis.
11. Fig. 10 examines effects of CREB activation and inhibition in vivo. The authors should address the physiological significance of this pathway after ADT by carrying out a similar study of LNCaP xenografts after castration (and ideally extend this to at least one other model).

Reviewers' comments:

We thank the reviewers for their efforts in reviewing our manuscript. We are truly grateful for their constructive suggestions, which makes our paper more impactful. We have worked very hard to address their comments/suggestions. Please see below our responses to their review summaries and individual comments. We have used a blue font color like this to distinguish the major new texts in the revision main text from the previous texts in our initial submission.

Reviewer #1 (Remarks to the Author):

In the manuscript titled, "Androgen deprivation promotes neuroendocrine differentiation and angiogenesis through CREB-EZH2-TSP1 pathway in prostate cancers," Zhang Y et al explore connections between CREB activation, EZH2 and thrombospondin in the setting of neuroendocrine prostate cancer models and tissues. While many of the results are interesting, the study appears to be too broad in scope at the expense of providing sufficient foundational evidence to support critical aspects of the processes that are investigated.

Response to review summary: To strengthen the paper, we have performed additional experiments and provided more foundational evidences. Major new experiments and their results include: (1) similar to CSS treatment in AR-positive cells, ADT drug enzalutamide (MDV3100) treatment regulates this pathway, i.e. inducing p-CREB/H3K27me3/NE and repressing TSP1, which is reversed by adding androgen DHT (Fig. 1a, 2a, 7a, 7b, 8a). All old data related to CSS treatment in AR-low PC3 cells have been removed. (2) Similar to MDV3100 *in vitro*, castration of NOD/SCID mice carrying LNCaP xenografts regulates the pathway, inducing NE marker expression and angiogenesis marker CD31 expression (Fig. 10e-f). (3) The effects of ADT can be reversed by inhibitors of EZH2 or PKA/CREB signaling *in vitro* (MDV3100, Fig. 1e, 1f, 3a, 3f, 8c, 8j, Supplementary Fig. S2) and *in vivo* (castration, Fig. 10e-f). (4) Re-expressing EZH2 cDNA reverses the induction of TSP1 and reduction of NE markers by EZH2 inhibition (Fig. 5f); (5) Silencing TSP1 induces NE marker expression and rescues the inhibition of angiogenesis by CREB signaling antagonist (Fig. 9e-f). (6) Concordant expression correlations have been shown in several additional large expression profiles of cancer patient samples (Fig. 6d, 6e, Supplementary Fig. S3 and S4).

1. Critical experiments to causally support their proposed mechanisms, as opposed to potential confounding experimental factors, are often missing. For example, in key gain- and loss-of-function experiments (e.g. Fig 4a,c; 5a,b) carrying out rescues with wild-type and mutant proteins would be important to demonstrate that the observations are causally connected.

Response: We have performed the suggested rescue experiments and presented the results in Fig. 4d and 5f, which supports the causal relationship between EZH2 and NE markers ENO2 and CHGA (Fig. 4d), and between EZH2 and TSP1 (Fig. 5f). We showed that silencing EZH2 using shRNA reduces NE marker expression, which is rescued by expressing a Doxycycline inducible EZH2 cDNA (Fig. 4d). Similarly, silencing EZH2 by siRNA upregulates TSP1, which is

reversed by re-expressing of siRNA-resistant EZH2 cDNA (Fig. 5f). Together with other data in Fig. 4 and 5, we believe that our conclusions regarding EZH2 induction of NE markers and EZH2 repression of TSP1 are well-supported.

2. Related to the above, in Figure 1A, 2A, 7A: the results in PC3 are confusing. The increase in p-CREB, H3K27me3 and decrease in TSP1, in PC3 cells is not likely to be due to androgen deprivation since these cells do not express AR. Rather, charcoal stripped serum is depleted of many factors in addition to depletion of androgens, and it is likely that one of these other factors may be responsible for the differences in the CSS vs. FBS conditions. This then calls to question whether the results in LNCaP cells are similarly due to depletion of some other factor other than androgen receptor ligands. In these experiments, rescue with androgen (DHT, or R1881) with and without modulation of AR in the CSS cells would be critical to carry out.

Response: We agree with Reviewer-1's concerns about results from PC3 cells in medium with charcoal stripped serum (CSS). PC3 cells are known to express very low level of AR, although AR expression could be detected by real time PCR with high Cq values. To avoid confusion to the readers, we have removed all data related to PC3 cells in CSS medium. Instead, to mimic ADT in patients, we treated AR-positive cells, such as LNCaP, VCaP and 22Rv1, with AR antagonist Enzalutamide (MDV3100). We have shown that MDV3100 activates the p-CREB/H3K27me3/NE pathway and represses TSP1 expression, similar to CSS treatment of LNCaP cells (Fig. 1a, 2a, 7a, 7b, 8a). Importantly, these MDV3100 effects can be reversed by androgen (DHT) or by inhibitors to p-CREB/EZH2 signaling axis (Fig. 3f, Supplementary Fig. S2). These critical new data provide strong supports to the overall conclusion of this paper.

Reviewer #2 (Remarks to the Author):

Previous studies have found that ADT can activate CREB, and that this can promote neuroendocrine (NE) differentiation in prostate cancer (PCa) cells. Increased EZH2 also has been linked directly to NE differentiation in PCa cells. This study provides evidence indicating that CREB activation acts by increasing EZH2 activity. EZH2 has also been shown to be pro-angiogenic, at least in part by suppressing expression of the anti-angiogenic protein TSP1. Consistent with these data, this study provides evidence that CREB acts through EZH2 to suppress TSP1. While aspects of this study are largely confirmatory of previous data (such as the link between EZH2 and NE differentiation and angiogenesis), CREB regulation of EZH2 appears to be a novel finding. However, there are no data addressing the mechanism by which CREB activates EZH2, and such mechanistic data would clearly strengthen the conclusions. Moreover, while this study and previous data indicate that ADT can activate CREB (although there are concerns about effects seen in PC3 cells, see below), more data are needed to conclude that this activation (CREB) in response to ADT is indeed a major pathway driving EZH2 activation and NE differentiation. As noted below, this would include examining effects of castration *in vivo*, ideally in a model in addition to LNCaP. Further specific issues are detailed below.

Response to review summary: Please refer to the above lists of major new data we have obtained for this revision at the beginning of our responses to Reviewer-1's comments. In addition, as suggested by Reviewer-2, to strengthen the conclusion that CREB activation in response to ADT is a major pathway driving EZH2 activation and NE differentiation *in vitro* and *in vivo*, we have carried out epistasis experiments, by repressing CREB upon ADT and then checking changes in EZH2 activities and NE marker expression. Besides inhibitors for CREB signaling pathway, we utilized two different genetic approaches to directly repress CREB in LNCaP cells upon MDV3100 treatment *in vitro* and castration *in vivo*, i.e. Doxycycline inducible expression of either shCREB or ACREB (a dominant-negative CREB polypeptide). The new *in vitro* and *in vivo* data presented in Fig. 1e-f, 8c, 10e-f provide strong evidences that CREB signaling is critical for ADT activation of EZH2 and NE phenotypes, as detailed below.

The detailed mechanism underlying EZH2 activation by CREB signaling is currently unclear and under investigation. Since the scope of this paper is already broad, as pointed out by Reviewer-1, we did not address this part. In the revised discussion section, we postulate a potential mechanism of CREB activating EZH2. We previously reported that CREB activation induces HDAC2 expression (Hulsurkar M *et al*, *Oncogene* 2016). Since HDAC2 is known to cooperate with EZH2 in epigenetic regulation, we propose that induction of HDAC2 by CREB signaling may contribute to EZH2 activation by CREB signaling. We provided some preliminary data supporting this potential mechanism. As shown in Supplementary Fig. 2d, treatment of HDAC inhibitor TSA reverses the effects of MDV3100 in activating the p-CREB/EZH2/NE pathway and repressing TSP1.

1. In Fig. 1a should show in parallel effects on NE markers. More importantly, AR expression in PC3 cells is low or undetectable and they do not respond to androgen. Therefore, there are concerns that the effect of culturing in CSS medium is unrelated to AR.

Response: As explained above in addressing similar concern from Reviewer-1, we have removed all data related to PC3 in CSS media. In new Fig. 1a, we showed a parallel elevation of p-CREB and NE markers upon MDV3100 treatment in AR-positive LNCaP and VCaP cells. Similar parallel increases have also been observed in other MDV3100 experiments for LNCaP and 22Rv1, such as Fig. 2a. We also demonstrated that in AR-positive LNCaP and 22Rv1 cells, the effects of CSS are similar to MDV3100 treatment in activating the CREB/EZH2 axis, which is reversed by DHT or inhibitors to the axis (Fig. 1e, 2a and Supplementary Fig. S2).

2. Fig. 1d should address whether CREB depletion with several independent RNAi (or with rescue controls) decrease NE markers. Also, do the drugs used prevent the upregulation of NE markers after ADT?

Response: We have showed that *in vitro* CREB depletion, by two different approaches (shCREB and ACREB), decreases NE markers and prevents the upregulation of NE markers after MDV3100 treatment (Fig. 1e, 1f, 8c). Similar results were obtained in mouse xenografts of LNCaP cells. While castration *in vivo* increases CREB activation and NE markers in LNCaP xenografts, doxycycline-induced CREB silencing prevents the castration induction of H3K27me3

and NE markers. *In vivo* silencing of CREB also inhibits LNCaP xenograft growth in castrated NOD/SCID mice.

3. There are similar concerns regarding lack of AR in PC3 cells in Fig. 2a.

Response: Please refer to our responses to Reviewer-2's Comment #1 and Reviewer-1's Comment #2.

4. Given concerns about whether CSS is acting just via AR, Fig. 3e should further address whether addition of a direct AR antagonist (enzalutamide) stimulates CREB and H3K27me, and whether this is blocked by CREB and EZH2 inhibitors. The results should also be generalized to at least one other AR responsive PCa cell.

Response: Please see our response to Reviewer-1's Comment #1. Indeed, enzalutamide (MDV3100) stimulates p-CREB and H3K27me3, which is blocked by CREB signaling inhibitor propranolol (PRO), EZH2 inhibitor GSK126 or EPZ-6438 in multiple AR-positive prostate cancer cell lines, including LNCaP, 22Rv1 (Fig. 3f, Supplementary Fig. S2a-c).

5. Fig. 4e should show effects of GSK126 alone to determine whether FSK has an effect when EZH2 is blocked. This should also ideally be done at shorter times. As FSK rapidly activates CREB, the authors should establish a time course for increased H3K27me and NE markers, and look at effects of EZH2 inhibition at the earliest times an effect is seen.

Response: Following this suggestion, we have showed that GSK126 alone reduces H3K27me3 and NE marker CHGA (Supplementary Fig. 2c). We also performed the time course experiment with FSK treatments with and without GSK126. We found that FSK induces H3K27me3 and NE marker protein CHGA by 8 hr, which is reversed by GSK126 (Supplementary Fig. 2c).

6. There are concerns that as FSK and other agents used are such potent and rapid inducers of CREB activity, the effects seen may not be physiological. What is the time course for CREB stimulation, H3K27me, and induction of NE markers in LNCaP cells after treatment with enzalutamide, and what is the effect of CREB inhibition?

Response: Enzalutamide generally takes longer time to induce NE markers than direct activation of PKA/CREB signaling with FSK. Following this suggestion, we treated LNCaP cells with Enzalutamide (MDV3100) for 24 hr and 72 hr without or with Dox-induction of CREB inhibitory polypeptide ACREB. We observed that CREB activation is increased at 24 hr, less obviously induced at 72 hr, which is consistent with CREB's proposed role of an early responder to ADT. NE markers are readily induced by MDV3100 at 24 hr, which is maintained through the 72 hr treatment period. ACREB induction by Dox treatment can repress CHGA (not so much for ENO2) at 24 hr. At 72 hr, both ENO2 and CHGA are clearly reduced by Dox treatment.

7. Related to the above point and figure 5, does enzalutamide increase TSP1?

Response: We assumed that the reviewer meant to ask whether enzalutamide decreases TSP1, as expected from our data. Indeed, it does reduce TSP1 expression because it activates the p-CREB/EZH2 axis that represses TSP1 (Fig. 7b, 8a, 8c, 8j).

8. In Fig. 6, the authors should assess for EZH2 and TSP1 correlation in the much larger TCGA and SU2C data sets.

Response: The two datasets we had in our initial submission are well-cited prostate cancer datasets with some metastatic CRPC or NEPC samples (Beltran H *et al*, *Nat Med* 2016 and Grasso CS *et al*, *Nature* 2012), and thus are highly relevant to this study. TCGA prostate cancers are all primary tumors. Following this suggestion, we have examined additional larger human cancer datasets through www.cBioPortal.org interface. We found that EZH2 and TSP1 expression are negatively correlated in several TCGA solid cancer types and in another well-cited metastatic prostate cancer dataset (FHCRC, Kumar A *et al*, *Nat Med* 2016) (Fig. 6, Supplementary Fig. S3 and S4). In addition, we show that TSP1 negatively correlates, while EZH2 positively correlates, with NE markers in TCGA prostate cancer dataset and SU2C as well as FHCRC prostate cancer datasets (Supplementary Fig. S3 and S4).

9. In Fig. 8f, effects of ISO in PC3 seem very modest. How does this compare with effects on TSP1 mRNA?

Response: We have performed RT-PCR of TSP1 in ISO +/- PRO treated PC3 and LNCaP cells. Consistent with a more modest increase of ISO on H3K27me3 marks on TSP1 promoter in PC3 than in LNCaP cells, the repression of TSP1 expression by ISO is also more modest in PC3 cells (Fig. 8i).

10. It cannot be concluded from Fig. 9 that the effects seen are due to repression of TSP1. The more conclusive study would be to RNAi or CRISPR deplete TSP1 and then show that CREB activation or repression no longer affects angiogenesis.

Response: As suggested, we have performed this experiment and presented this critical new piece of data in Fig. 9e and 9f. The result indicates that TSP1 silencing rescues the angiogenesis inhibition by CREB repression, which, together with our other data, suggests that TSP1 induction by CREB repression is a critical mechanism for CREB's regulation of angiogenesis. Interestingly, we further observed that TSP1 silencing induces NE markers in PC3 cells, which further supports an intimate connection between angiogenesis and neuroendocrine phenotype that we have uncovered in this study.

11. Fig. 10 examines effects of CREB activation and inhibition *in vivo*. The authors should address the physiological significance of this pathway after ADT by carrying out a similar study of LNCaP xenografts after castration (and ideally extend this to at least one other model).

Response: Following this excellent suggestion, we have carried out castration experiment to test the physiological significance of this pathway after ADT. NOD/SCID mice carrying LNCaP-Dox-ACREB xenograft tumors were either uncastrated and fed with regular chow, or castrated and fed with regular chow (mimicking ADT), or castrated and fed with Dox-containing chow that induces ACREB, a dominant negative polypeptide inhibiting CREB signaling. Our new results from this experiment demonstrated that castration regulates the CREB/EZH2/NE/TSP1 pathways in a concordant manner as our *in vitro* results. Importantly, repressing CREB *in vivo* reverses the effects of castration and inhibits tumor growth in castrated mice. These critical pieces of new data provide further strong supports for the physiological relevance of our study and the novel pathway we have identified for prostate cancer progression.

We did try to expand this *in vivo* experiment to another AR-positive prostate cell line. A well-known problem in the prostate cancer research field is that the number of suitable cell models is quite limited. We performed this experiment using AR-high VCaP cells. However, with AR amplification and splicing variants, VCaP xenografts grow much faster than LNCaP xenografts and only responded very briefly to castration. We had to sacrifice the mice prematurely, because of tumor sizes.

Reviewers' Comments:

Reviewer #1:

Remarks to the Author:

The authors have carried out significant new experiments and have addressed many of my concerns. However, the causal connection between androgen deprivation and AR-antagonist treatment and induction of p-CREB/H3K27me3/NE pathway induction is still not well established in the new experiments. In the revised Figures 1A, 2A, the induction of p-CREB and H3K27me3 by treatment with MDV3100 does not appear to be rescued by addition of DHT. It is also confusing why the authors switched to this AR-antagonist system instead of rescuing the CSS conditions with DHT. If the entire p-CREB/H3K27me3/NE program is not rescued, but only TSP1 loss is rescued, this is fine – the authors should just discuss potential reasons for this.

Reviewer #2:

Remarks to the Author:

The authors have added substantial additional data that have adequately addressed the issues raised in the initial review.

REVIEWERS' COMMENTS:

Reviewer #1 (Remarks to the Author):

The authors have carried out significant new experiments and have addressed many of my concerns. However, the causal connection between androgen deprivation and AR-antagonist treatment and induction of p-CREB/H3K27me3/NE pathway induction is still not well established in the new experiments. In the revised Figures 1A, 2A, the induction of p-CREB and H3K27me3 by treatment with MDV3100 does not appear to be rescued by addition of DHT. It is also confusing why the authors switched to this AR-antagonist system instead of rescuing the CSS conditions with DHT. If the entire p-CREB/H3K27me3/NE program is not rescued, but only TSP1 loss is rescued, this is fine – the authors should just discuss potential reasons for this.

Response to review summary: Many thanks to Reviewer #1 for being satisfied with our significant new data and the revised manuscript.

Regarding DHT rescue in Fig. 1a and 2a, we believe that DHT does rescue (i.e. reverse), to a good extent, although not completely, the MDV3100 impacts on the p-CREB/H3K27me3/NE pathway. For LNCaP cells in previous Fig 1a, if we had a weaker exposure of the p-CREB western blot film, the rescue of p-CREB may be more obvious. p-CREB western blot signal was strong. We have repeated the LNCaP western blot with weaker exposure and provided a revised Fig. 1a, where it is obvious that DHT rescues, at least partially, the p-CREB/NE pathway. The p-CREB/NE rescue by DHT is also evidenced for VCaP cells in the revised Fig. 1a.

We have also provided a revised Fig. 2a, where a new western blot for LNCaP cells was performed with optimal film exposure. The H3K27me3 band for 22RV1 in Fig. 2a has been replaced with a weaker exposure band from the same 22RV1 western blot. It is clear that elevated levels of H3K27me3 and NE markers by MDV3100 treatment are rescued/reversed by DHT. The reason that DHT can not completely rescue the MDV3100 effects may be because MDV3100 can still, in some degree, inhibit AR activity with the addition of DHT, by competing with DHT for AR binding. Therefore, it may be a dose matter.

We switched from CSS to MDV3100 as the main ADT method in last revision, because MDV3100 treatment more closely mimics the ADT in prostate cancer patients than CSS. It was also because there were concerns raised previously by Reviewer #1 and #2 that CSS is depleted of many factors in addition to androgen, therefore it was unclear to us how to interpret results of DHT rescue of the CSS effects. In Fig. 2a, we showed a DHT rescue of H3K27me3/NE induction by CSS+MDV3100 in AR-positive 22RV1 cells.

Reviewer #2 (Remarks to the Author):

The authors have added substantial additional data that have adequately addressed the issues raised in the initial review.

Response to review summary: Many thanks to Reviewer #2 for being very satisfied with our significant new data and the revised manuscript.